

# Modelling carbonaceous aerosol from residential solid fuel burning with different assumptions for emissions

Riinu Ots[1,2,*], Mathew R. Heal[1], Dominique E. Young[3,**], Leah R. Williams[4], James D. Allan[3,5],
Eiko Nemitz[2], Chiara Di Marco[2], Anais Detournay[2], Lu Xu[6,***], Nga L. Ng[6,7], Hugh Coe[3],
Scott C. Herndon[4], Ian A. Mackenzie[8], David C. Green[9], Jeroen J. P. Kuenen[10], Stefan Reis[2,11], and
Massimo Vieno[2]

[1]School of Chemistry, University of Edinburgh, Edinburgh, UK
[2]Natural Environment Research Council, Centre for Ecology & Hydrology, Penicuik, UK
[3]School of Earth, Atmospheric and Environmental Sciences, University of Manchester, Manchester, UK
[4]Aerodyne Research, Inc., Billerica, MA, USA
[5]National Centre for Atmospheric Science, University of Manchester, Manchester, UK
[6]School of Chemical and Biomolecular Engineering, Georgia Institute of Technology, Atlanta, GA, USA
[7]School of Earth and Atmospheric Sciences, Georgia Institute of Technology, Atlanta, GA, USA
[8]School of GeoSciences, University of Edinburgh, Edinburgh, UK
[9]MRC PHE Centre for Environment and Health, King's College London, London, UK
[10]TNO, Department of Climate, Air and Sustainability, Utrecht, The Netherlands
[11]University of Exeter Medical School, European Centre for Environment and Health, Knowledge Spa, Truro, UK
[*]now at: Clinical Surgery, University of Edinburgh, Edinburgh, UK
[**]now at: Air Quality Research Center, University of California, Davis, CA, USA
[***]now at: Division of Geological and Planetary Sciences, California Institute of Technology, Pasadena, CA, USA.

*Correspondence to:* M. Heal (M.Heal@ed.ac.uk) and R. Ots (R.Ots@ed.ac.uk)

**Abstract.** Evidence is accumulating that emissions of primary particulate matter (PM) from residential solid fuel combustion in the UK may be underestimated and/or spatially misclassified. In this study, different assumptions for the spatial distribution and total emission of PM from solid fuel (wood and coal) burning in the UK were tested using an atmospheric chemical transport model. Modelled concentrations of the PM components were compared with measurements from aerosol mass spectrometers at four sites in central and Greater London (ClearfLo campaign, 2012), as well as with measurements from the UK black carbon network. The two main alternative emission scenarios modelled were *Base4x* and *combRedist*. For *Base4x*, officially reported $PM_{2.5}$ from the residential and other non-industrial combustion source sector were increased by a factor of 4. For the *combRedist* experiment, half of the baseline emissions from this same source were redistributed by residential population density to simulate the effect of allocating some emissions to the smoke control areas (that are assumed in the national inventory to have no emissions from this source). The *Base4x* scenario yielded better daily and hourly correlations with measurements than the *combRedist* scenario for year-long comparisons of the solid fuel organic aerosol (SFOA) component at the two London sites, whereas the latter scenario better captured mean measured concentrations across all four sites. The modelled elemental carbon (EC) concentrations derived from the *combRedist* experiments also compared well with seasonal-average concentrations of black carbon observed across the network of UK sites. Together, the two model scenario simulations of SFOA and EC suggest both that residential solid-fuel emissions may be higher than inventory estimates and that the spatial distribution of





residential solid-fuel burning emissions, particularly in smoke control areas, needs re-evaluation. The model results also suggest the assumed temporal profiles for residential emissions may require review to place greater emphasis on evening (including 'discretionary') solid-fuel burning.

## 1 Introduction

Globally, solid fuel burning from residential heating and from cooking activities is a major source of both indoor and outdoor $PM_{2.5}$ (particulate matter with diameter $< 2.5\,\mu m$) air pollution (WHO, 2015). As developed countries commit to renewable energy targets, the use of wood and biomass in residential heating is likely to increase (replacing some of the natural gas based heating systems (WHO, 2015). Within Europe, residential wood burning is estimated to be the single largest anthropogenic primary source of organic carbon (OC), contributing ~60% of total OC emissions from European countries (Denier van der Gon et al., 2015; Bergström, 2015). In some countries, both wood and coal are burned in residential stoves and other small combustion plants. A number of particle source apportionment studies, particularly those based on positive matrix factorisation of aerosol mass spectrometry measurements (AMS-PMF studies), have thus attributed organic aerosol (OA) from this source as solid fuel OA (SFOA), which is a combination of the commonly known biomass burning OA (BBOA) factor plus coal burning OA (Allan et al., 2010; Young et al., 2015a; Xu et al., 2016). In addition to heating, some residential solid fuel burning in European urban areas can also be attributed to recreation (i.e. fireplaces for ambience Fuller et al. (2013)).

Since the Great London Smog of 1952, several legislative interventions have substantially reduced the use of solid fuels for residential heating in the UK by subsidising infrastructure and availability of oil and natural gas, as well as the implementation of smoke control areas (Fuller et al., 2013). For example, almost all of London is now a smoke control area where solid fuel burning is prohibited unless undertaken in approved wood burners (Fuller et al., 2013). This control is applied only to appliances with a chimney; incidental sources such as bonfires or barbecues are permitted.

There is evidence, however, that the smoke control legislation is no longer actively enforced. Several recent studies have reported substantial local contributions of emissions from solid fuel burning to particle concentrations in London coinciding with days of low temperature (Fuller et al., 2014; Crilley et al., 2015). This is relevant as, currently, the UK's National Atmospheric Emissions Inventory (NAEI) assumes zero residential emissions of non-approved solid fuel burning in smoke control areas (i.e. that there is full legal compliance). Furthermore, the NAEI only includes estimates of emissions from officially-sold solid fuels (NAEI, 2013), but there is reason to believe that much fuel wood is not obtained through commercial outlets and falls outside the economic administration and therefore is not included in official statistics (Denier van der Gon et al., 2015). For example, a recent UK Wood Use Survey concluded that the official national consumption of domestic wood fuel is underestimated by a factor of 3 (Waters, 2016). Furthermore, the survey showed that 31% of wood fuel is sourced from the informal "grey" market of wood (i.e. own garden, other landowners' gardens, waste wood, etc.).

The results of the UK survey are in line with recent evaluations of wood burning emissions in Belgium, which also concluded that their official inventory has underestimated the amount of wood burned in residential settings by more than a factor of 3 (Lefebvre et al., 2016). Belgium has since included these increased emissions estimates (adding estimates for non-commercial



wood sources to officially reported sales) in the amount reported to the Centre on Emission Inventories and Projections (CEIP) of the European Monitoring and Evaluation Programme (EMEP). The authors of the Belgian study urge countries that currently do not make estimates of non-officially traded or sourced wood to follow the same practise and start reporting these.

Model-measurement comparisons of a range of gaseous and particulate pollutants (Ots et al., 2016a) show that modelled concentrations of SFOA at the North Kensington site in London are substantially underestimated (normalised mean bias, NMB, of $-71\%$) compared to an annual dataset of PMF apportionment of AMS measurements collected during the 2012 Clean Air for London campaign (ClearfLo; Bohnenstengel et al. (2014); Young et al. (2015a); Xu et al. (2016)). The aim of this work is to use these measurements of SFOA as the basis for an atmospheric chemical transport model exploration of potential closure of this discrepancy.

## 2    Methods

### 2.1    Model description

EMEP4UK is a regional application of the EMEP MSC-W (European Monitoring and Evaluation Programme Meteorological Synthesizing Centre-West) chemical transport model (Vieno et al., 2009, 2010, 2016; Ots et al., 2016a, b). The EMEP MSC-W model is a 3-D Eulerian model that has been used for both scientific studies and to support policy making in Europe. A detailed description of the EMEP MSC-W model, including references to evaluation and application studies is available in Simpson et al. (2012), Schulz et al. (2013), and at www.emep.int. The model configuration used here was based on version v4.5. The model has 21 vertical layers, extending from the surface to 100 hPa. The lowest vertical layer is ~40 m in height, and the horizontal resolution over an inner domain covering the British Isles is $5\,\mathrm{km} \times 5\,\mathrm{km}$. The model uses one-way nesting from an extended European domain (simulated with $50\,\mathrm{km} \times 50\,\mathrm{km}$ horizontal resolution). The model was driven by output from the Weather Research and Forecasting (WRF) Open Source model (www.wrf-model.org, version 3.1.1) including data assimilation of 6-hourly meteorological reanalysis from the US National Center for Environmental Prediction (NCEP)/National Center for Atmospheric Research (NCAR) Global Forecast System (GFS) at $1°$ (~100 km) resolution (NCEP, 2000).

Gridded anthropogenic emissions were obtained from NAEI (NAEI, 2013) for the UK, and from CEIP (CEIP, 2015) for the rest of Europe. All emissions are apportioned across a standard set of emission source sectors, following the sector structure defined in the Selected Nomenclature for Air Pollutant (SNAP, Table 1; EEA (2013)). Daily emissions from natural fires were taken from the Fire INventory from NCAR version 1.0 (FINNv1, Wiedinmyer et al. (2011)). Primary PM emissions reported as $PM_{2.5}$ and $PM_{10}$ in NAEI and CEIP were speciated into elemental carbon (EC), hydrocarbon-like OA (HOA) from fossil fuel combustion, OA from domestic combustion (SFOA = BBOA + coal OA) and remaining primary PM for each source sector using splits developed by Kuenen et al. (2014), as shown in Fig. 1. In all the experiments presented here, SFOA is assumed to be non-volatile and it does not undergo atmospheric ageing.

The performance of this version of the EMEP4UK model simulating a standard suite of gas-phase components and secondary inorganic aerosol PM components is reported in Ots et al. (2016a) comparing with a full year of measurements in London in 2012. In brief, Ots et al. (2016a) reported an NMB of $-1\%$ and $r = 0.79$ for ozone, an NMB of $-32\%$ and $r = 0.78$ for $NO_x$,





**Table 1.** SNAP source sectors as specified in the emissions input to the model (CEIP, 2015).

| SNAP1 | Combustion in energy and transformation industries |
| --- | --- |
| SNAP2 | Residential and non-industrial combustion |
| SNAP3 | Combustion in manufacturing industry |
| SNAP4 | Production processes |
| SNAP5 | Extraction and distribution of fossil fuels |
| SNAP6 | Solvent and other product use |
| SNAP7 | Road transport |
| SNAP8 | Other mobile sources and machinery |
| SNAP9 | Waste treatment and disposal |
| SNAP10 | Agriculture |

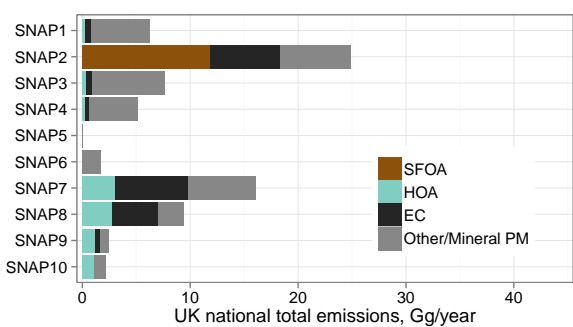

**Figure 1.** Annual UK $PM_{2.5}$ emissions by SNAP sector as specified in the NAEI (for year 2012), with each sector split into primary OA (HOA or SFOA), EC, and remaining PM following Kuenen et al. (2014). Source: Ots et al. (2016a).

an NMB of $+6\%$ and $r = 0.73$ for $SO_4$, an NMB of $-12\%$ and $r = 0.65$ for $NH_4$, and an NMB of $-23\%$ and $r = 0.57$ for $NO_3$.

## 2.2 Model experiments

In this study, four different cases were considered. The *Base* case model experiment uses the same emission inventory dataset
as Ots et al. (2016a) (i.e. as reported by the NAEI using the splits in Fig. 1), but with a small adjustment in the daily variation in
emissions due to temperature, called degree-day factors (Simpson et al., 2012). Degree-day factors modulate the daily variation
in emissions from the SNAP2 sector according to ambient temperature (i.e. increasing the emissions during colder days).
SNAP2 includes $PM_{2.5}$ emissions from both residential and small (non-industrial) commercial combustion, but the residential
part dominates with annual emissions for 2012 of 19 Gg from residential, 1 Gg from commercial and 5 Gg from stationary





**Table 2.** Summary of the four experiments for $PM_{2.5}$ emissions from SNAP2. In all experiments, $PM_{2.5}$ is split into three components as follows: 48% is SFOA, 26% EC, and 26% other/mineral PM (as in Fig. 1).

| Experiment | SNAP2 $PM_{2.5}$ emission | Spatial distribution |
|---|---|---|
| *Base* | 25 Gg | NAEI |
| *Base4x* | 100 Gg | NAEI |
| *Redist* | 25 Gg | Population density |
| *combRedist* | 25 Gg | NAEI + Population density |

military combustion (NAEI, 2013). The domination of the residential emissions means that no large additional uncertainty is introduced by applying the degree-day factors to the whole of SNAP2.

A degree-day is defined as $H_{dd,j} = max(18°C - T_j^{24h}, 1)$, where $j$ is the day number and $T^{24h}$ is the daily averaged tem-perature in $°C$. These degree-days are divided by the annual mean $(\overline{H_{jj}})$ to obtain degree-day factors (Simpson et al., 2012).

For this work, degree-day factors pre-calculated by the EMEP MSC-W Centre based on European Centre for Medium-Range Weather Forecasts (ECMWF) 2012 meteorological simulations for the 50 km × 50 km domain (Schulz et al., 2013) were disaggregated using the simple area-weighting method assuming homogeneity for degree-day factors within the 50 km to 5 km conversion.

In the second model experiment (*Base4x*), emissions from SNAP2 were increased by a factor of 4 (based on the NMB of
−71% at the London North Kensington site). SNAP2 is the only sector with SFOA emissions in the set-up used (Fig. 1). The spatial distribution was unchanged.

For the third experiment (*Redist*), the annual reported $PM_{2.5}$ emissions from SNAP2 were re-gridded linearly by residential population density (census data from Reis et al. (2016)) in order to test the assumption made by the NAEI that only smokeless fuels are used in smoke control areas. The total emission was unchanged.

Finally, the fourth experiment, *combRedist*, was a hybrid of the *Base* and *Redist* experiments, where, for each grid cell, half the emissions of *Base* and half of *Redist* (of the SNAP2 sector) were added together, i.e. $combRedist_{emis} = 0.5 \times Base_{emis} + 0.5 \times Redist_{emis}$. The experiments *Base*, *Redist*, and *combRedist* therefore use the same total emission; the only difference is in the spatial distribution of the emissions. All four experiments use the same temporal variation for SNAP2, including degree-day factors. The emissions used for the four experiments are summarised in Table 2. In all of these experiments, SFOA is assumed
to be non-volatile. The emission maps of these four experiments are shown in Fig. 2, and the resulting modelled annual average surface concentrations of SFOA are shown in Fig. 3.

## 2.3 Comparison with measurements

Modelled $OA_{2.5}$ (OA with diameter < 2.5 µm) is compared with non-refractory submicron (NR-PM$_1$) OA measured by Aerodyne AMS instruments at the London North Kensington and London Marylebone Road sites in central London (urban
background and roadside, respectively), and at the Harwell and Detling rural sites located to the west and east of London,



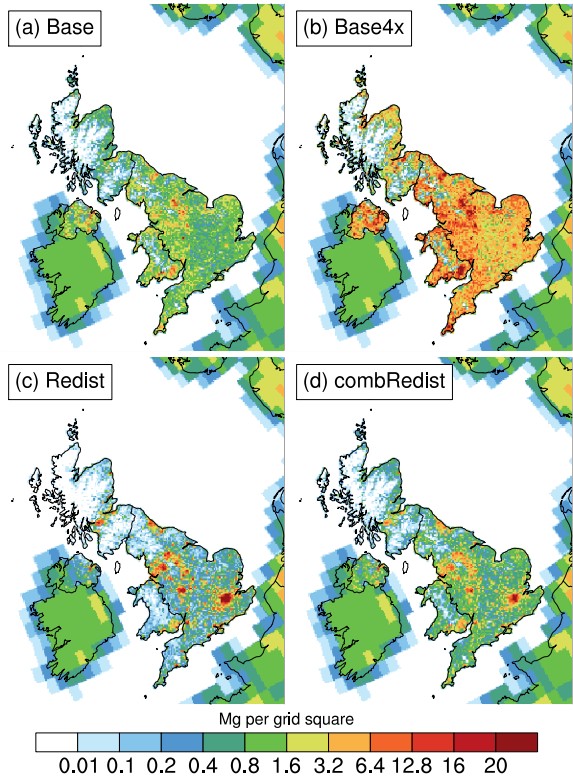

**Figure 2.** Total SFOA emissions (defined as 48% of $PM_{2.5}$ from SNAP2) for the year 2012 in the inner nesting domain for the four scenarios of this study: (a) *Base*: as in the NAEI, (b) *Base4x*: Base increased by a factor of 4 over the whole of UK, (c) *Redist*: UK emissions redistributed to residential population density (national total same as Base), (d) *combRedist*: half of the total emission redistributed to residential population density, half as reported by NAEI (a combination of (a) and (c)). The national total is the same for (a), (c), and (d). All UK emissions are aggregated to the 5 km × 5 km grid from an initial resolution of 1 km × 1 km reported by the NAEI. The emission resolution is 50 km × 50 km for other countries (as in CEIP). Note the non-linear scale.

respectively (Bohnenstengel et al., 2014; Young et al., 2015a, b; Xu et al., 2016). The locations of the AMS measurement sites used in this study are shown in Fig. 4. The discrepancy introduced by the different size fractions of modelled and measured concentrations is small as at an urban background site, ~90% of organic carbon in $PM_{2.5}$ is in the submicron fraction (Harrison and Yin, 2008).

5    Different types of AMS instruments were deployed in the ClearfLo campaign. At the London North Kensington site a compact time-of-flight AMS (cToF-AMS) was deployed for a full calendar year (January 2012 – January 2013), and an additional high-resolution time-of-flight AMS (HR-ToF-AMS, DeCarlo et al. (2006)) was deployed for the winter Intensive Observation Period (IOP). The measurements at Marylebone Road were taken with a Q-AMS (Quadrupole AMS; Jayne et al. (2000)). HR-ToF-AMS instruments were deployed at Detling and Harwell during the winter IOP (not the whole year). PMF analysis

10   was applied to each dataset to apportion measured OA into different components (Ulbrich et al., 2009). A detailed description





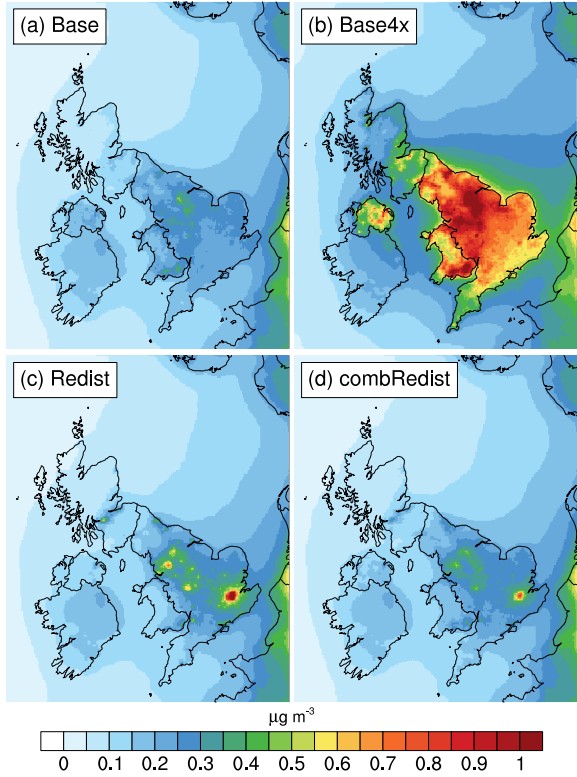

**Figure 3.** Annual average modelled surface SFOA concentrations for the year 2012 using the emission scenarios shown in Fig. 2: (a) *Base*: as in the NAEI, (b) *Base4x*: *Base* increased by a factor of 4 over the whole of UK, (c) *Redist*: UK emissions redistributed by residential population density (national total same as *Base*), (d) *combRedist*: half of the total emission redistributed by residential population density, half as reported by NAEI (a combination of (a) and (c)). The national total is the same for (a), (c), and (d).

of the derivation and optimization of the factors retrieved from the AMS data at Detling can be found in Xu et al. (2016), at London North Kensington in Young et al. (2015a) and Young et al. (2015b), at London Marylebone Road in Detournay et al. (2017) (all of these analyses were performed with the PMF2 solver), and at Harwell in Di Marco et al. (2017) (using the ME-2 solver). The limitations and uncertainties of these measurement datasets have been discussed in Ots et al. (2016a, b) .

5    The following numerical metrics were used for model evaluation: FAC2 (Factor of 2) - the proportion of modelled concentrations that are within a factor of 2 of the measured concentrations; NMB - normalised mean bias; NMGE - normalised mean gross error, which is defined as:

$$NMGE = \frac{\frac{1}{n}\sum_{i=1}^{n}|M_i - O_i|}{\overline{O}}, \tag{1}$$





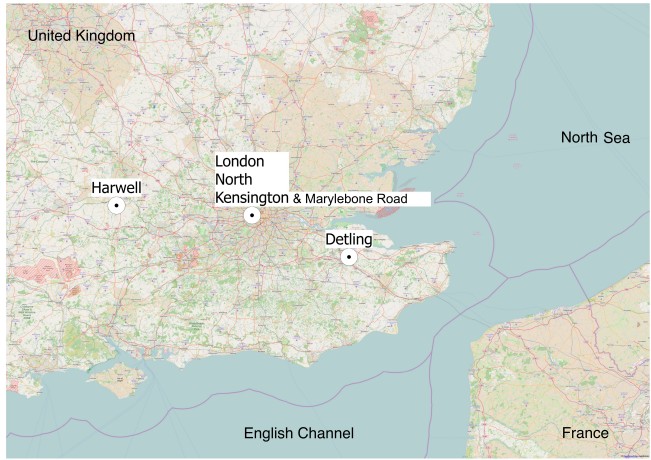

**Figure 4.** Locations of the ClearfLo measurement sites used in this work. London North Kensington is an urban background site, London Marylebone Road is a roadside site, Harwell and Detling are rural background sites (Source: Ots et al. (2016a)). Underlying map from © OpenStreetMap contributors.

where $M_i$ is the $i$th modelled value, $O_i$ is the corresponding measured value, $\overline{O}$ is the mean measured value, and $n$ is the total number of observations; $r$ - correlation coefficient; and COE - coefficient of efficiency, which is defined as:

$$COE = 1.0 - \frac{\sum_{i=1}^{n} |M_i - O_i|}{\sum_{i=1}^{n} |\overline{O} - O_i|}. \tag{2}$$

A COE of 1 indicates perfect agreement between model and measurements. Although the COE does not have a lower bound,
5 a zero or negative COE implies that the model cannot explain any of the variation in the observations (Legates and McCabe, 2013).

## 3 Results and Discussion

Figure 3 shows the annual mean modelled SFOA surface concentrations for the year 2012. In the model, SFOA is emitted as 48% of $PM_{2.5}$ from the SNAP2 source sector, and it is advected as a non-volatile and chemically inert species (but it is
10 included in the total OA budget for the absorptive partitioning of secondary organic aerosol species). The gradients of SFOA surface concentrations visible over the North Sea and the English Channel are indicators of European transport. Over the UK, SFOA concentrations follow the pattern of the prescribed local emissions, with the spatial distributions of the experiments with and without redistributed national emissions being substantially different from each other. The *Base* and *Base4x* scenarios (spatially gridded as reported by the NAEI) assigned most emissions to Northern England, Wales and Northern Ireland, leaving
15 major cities such as London, Manchester, Leeds, Birmingham, and Glasgow almost unnoticeable in the concentration fields (these, and many other urban locations, are designated as smoke control areas). In contrast, the *Redist* experiment highlights all of these urban areas, because the SFOA emissions were redistributed linearly by residential population density. The *combRe-*





*dist* experiment shows these residential hot-spots while also retaining some of the spatial pattern from the officially reported distribution.

## 3.1 Daily evaluation - London Marylebone Road and North Kensington annual datasets

Time-series of measured and modelled daily-average SFOA concentrations for the London Marylebone Road and North Kensington sites are shown in Fig. 5 (*Base*, and *Base4x*) and Fig. 6 (*Base*, and *combRedist*). These figures also highlight the date of the annual bonfire celebrations around Guy Fawkes Night (5th of November) to draw attention to an increase in SFOA emissions that can not be simulated with the model, given that the temporal variation is prescribed using a regular approach (i.e. hour-of-day, day-of-week, month-of-year) which does not include information about specific days and events. (The small difference between the *Base* experiment time-series in these figures and a similar comparison in Ots et al. (2016a) is due to the use of degree-day factors.) For convenience, the evaluation statistics presented on these time-series (as well as for the Redist experiment) are given in Table 3.

**Table 3.** Evaluation statistics for modelled vs measured daily-average concentrations of SFOA for the ClearfLo year-long datasets (year 2012).

| Site | Experiment | NMB | NMGE | $r$ | COE |
|---|---|---|---|---|---|
| | *Base* | -59% | 64% | 0.60 | -0.23 |
| | *Base4x* | 14% | 54% | 0.67 | -0.02 |
| Marylebone Road | *Redist* | 57% | 79% | 0.57 | -0.51 |
| | *combRedist* | -1% | 51% | 0.59 | 0.03 |
| | *Base* | -69% | 70% | 0.73 | 0.03 |
| | *Base4x* | -18% | 42% | 0.78 | 0.42 |
| North Kensington | *Redist* | 33% | 57% | 0.71 | 0.21 |
| | *combRedist* | -18% | 43% | 0.73 | 0.41 |

The experiments result in better daily-average SFOA model-measurements agreement than the *Base* case at the two sites. The only exception is the *Redist* simulation at Marylebone Road, where the underestimation of the *Base* case is replaced with an equivalent overestimation (NMB of $-59\%$ and $+57\%$, *Base* and *Redist*, respectively, Table 3). There is a small decrease in the *r*-value, and an increase of the NMGE, which is caused by the modelled values of *Redist* being greater than those of the *Base* experiment. For the London North Kensington site, the *Redist* experiment is an improvement compared to the *Base* run, although the concentrations are also overestimated (NMB = $+33\%$, *Redist*). This is expected, as areas with high population densities would include large apartment buildings which are unlikely to have individual fireplaces. Therefore a completely linear redistribution of residential emissions is not correct, but this experiment gives an indication of the maximum effect that population density could have on SFOA concentrations.




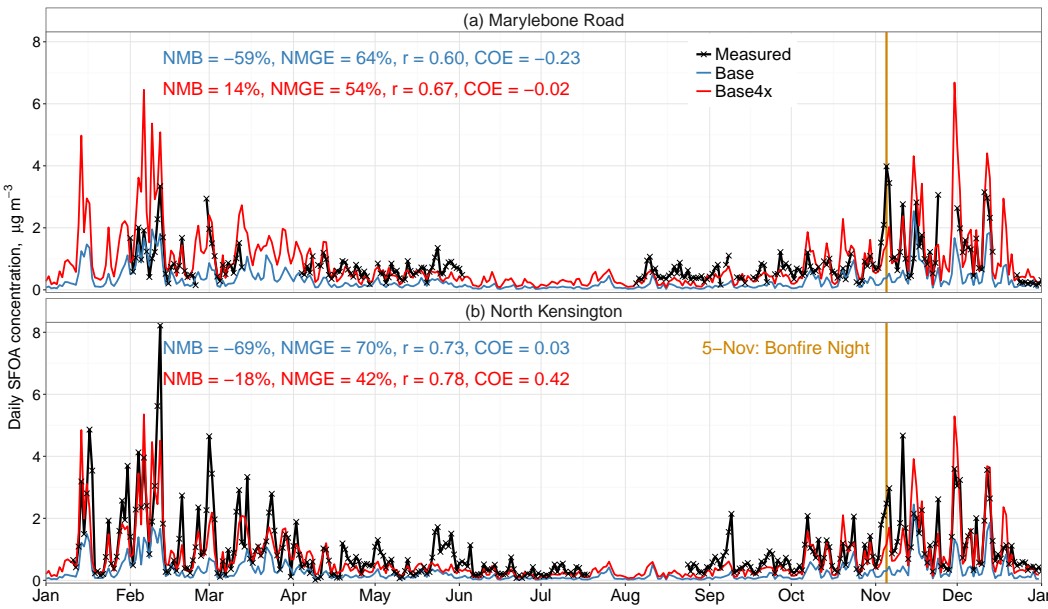

**Figure 5.** Time-series of measured and modelled (*Base* and *Base4x* experiments) daily-average SFOA concentrations at the (a) Marylebone Road, and (b) North Kensington measurement sites, year 2012. The vertical line marks the date of 'bonfire night', November 5th.

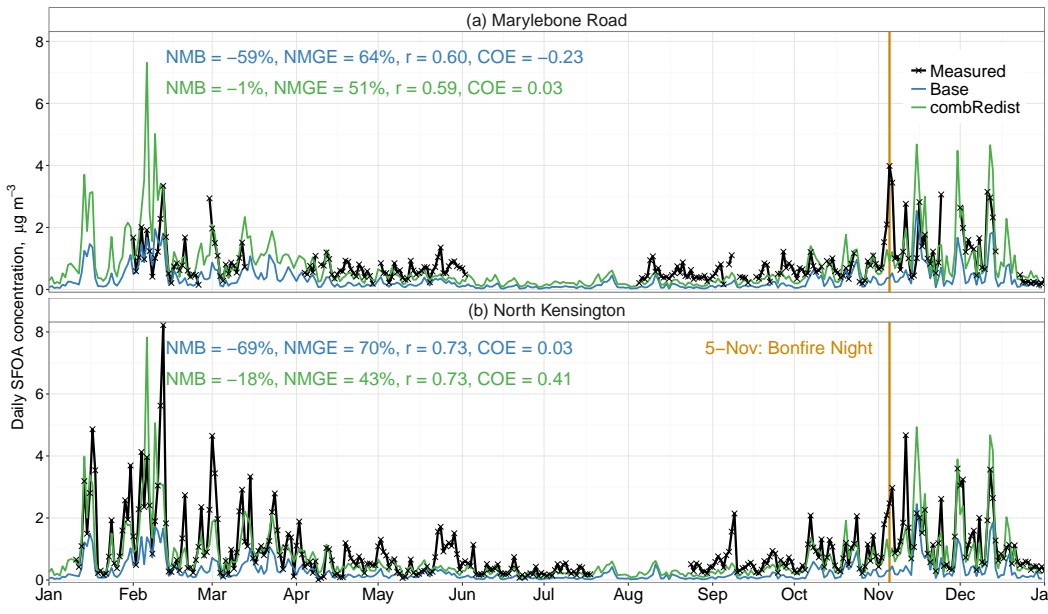

**Figure 6.** Time-series of measured and modelled (*Base* and *combRedist* experiments) daily-average SFOA concentrations at the (a) Marylebone Road, and (b) North Kensington measurement sites, year 2012. The vertical line marks the date of 'bonfire night', November 5th.



Both the *Base4x* and *combRedist* experiments have better predictive abilities for the AMS-PMF measured concentrations of SFOA at the two sites in London than the *Base* case emissions simulation. The NMGE at the Marylebone Road site is reduced to 54% (*Base4x*) or 51% (*combRedist*), compared to 64% for the *Base* case simulation. At the North Kensington site, NMGE is reduced to 42% (*Base4x*) or 43% (*combRedist*), compared to 70% for the *Base* case simulation. The *Base4x* results

in improvements in the *r*-value: 0.67 (*Base*: 0.60) and 0.78 (*Base*: 0.73) at Marylebone and North Kensington, respectively, whereas with the *combRedist* emissions, the *r*-values of daily-average concentrations remain the same as *Base*: 0.59 (0.60) and 0.73 (0.73), at Marylebone Road and North Kensington, respectively. The improvement in COE values is similar for both experiments (Table 3) at both sites. Both experiments decreased the NMB at North Kensington to $-18\%$ (*Base4x* and *combRedist*) from $-69\%$ (*Base*), whereas at Marylebone Road, the *Base4x* reaches an overestimation of NMB $= +14\%$ while

the *combRedist* matches the measured mean SFOA: NMB $= -1\%$ (NMB of *Base* at the Marylebone Road measurement site: $-59\%$). It should be noted, however, that there are several days in November and December where both the *Base4x* and *combRedist* experiments overestimate SFOA concentrations compared to measurements.

In summary, in comparison with daily-average measurements of SFOA concentrations at two sites in London the *Base4x* and the *combRedist* experiments resulted in similar improvements in NMGE and COE, the *Base4x* experiment had better *r*-

values, and the *combRedist* experiment better matched the annual mean concentrations of SFOA at the two sites. Nevertheless, it should be noted that AMS-PMF apportionment measurements are also subject to uncertainty which limits the expected correlation with the model (for further discussion on this see Ots et al. (2016a)).

The following sections evaluate these experiments with respect to hourly-average measurements taken with the High Resolution (HR-ToF-AMS) instruments during the ClearfLo winter IOP, which included two rural background sites - Harwell and

Detling - as well as the London North Kensington site.

### 3.2   Hourly averaged diurnal profiles of SFOA concentrations, winter IOP

Hourly averaged diurnal profiles of measured and modelled SFOA concentrations at the ClearfLo winter IOP sites are shown in Fig. 7. The profiles of measured concentrations at the rural background sites (Detling and Harwell), and the urban background site (North Kensington) all show a pronounced maximum in the evening (after 18:00), and a much smaller peak at around 9:00.

In the model, the hourly emission factors applied to the SNAP2 sector have similar magnitudes for morning and evening thus underestimating the higher evening concentrations seen in the measurements (in this work, the same diurnal emission profile was used for all countries). This temporal misclassification of emissions for the UK is expected to have a detrimental effect on all of the model hourly evaluation statistics, except NMB.

### 3.3   High SFOA episode: 13-Jan–18-Jan, 2012

Time-series of hourly-average measured and modelled (*Base*, *Base4x*, and *combRedist* experiments) SFOA concentrations at the ClearfLo winter IOP sites are shown in Fig. 8. Note the very high concentrations measured at Harwell from 13 to 18 Jan. North Kensington and Detling also exhibited elevated concentrations (although not as high as Harwell) on these days, especially on 17 Jan. During this episode, both the *Base4x* and *combRedist* experiments simulate similar concentrations for





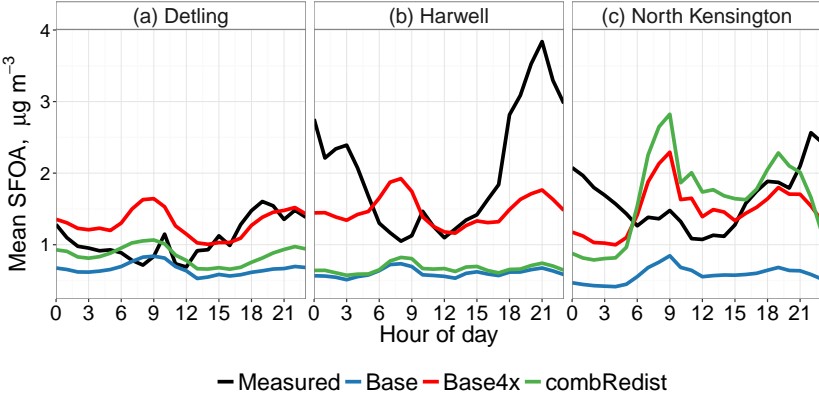

**Figure 7.** Hourly-averaged diurnal profiles of modelled and measured SFOA concentrations at the (a) Detling, (b) Harwell, and (c) North Kensington measurement sites, winter IOP 2012.

the North Kensington site, whereas for Harwell and Detling, the *Base4x* gives higher concentrations than *combRedist* and is therefore slightly closer to the very high measured concentrations. Note that *Base4x* only increased the UK emissions, not European ones. Hence the increase in *Base4x* concentrations compared with *Base* is not fourfold.

Daily-average maps of modelled SFOA surface concentrations during these days are shown in Fig. 9 (*Base4x*), and Fig. 10 (*combRedist*). Time-series of daily-average concentrations during the winter IOP are shown in Fig. 11 (using a threshold of 75% of valid hourly values to derive a measured daily average). Ots et al. (2016a) demonstrated that a rural background site can on occasion exhibit substantially higher concentrations than central London due to atmospheric import of polluted air masses from Europe creating a strong spatial gradient. The daily-average concentration maps, however, do not indicate European gradients over southern England during 13 to 18 Jan. Nevertheless, southern England did experience a sustained high-pressure weather system during these days. Sustained high pressure usually leads to a very stable atmosphere with descending air masses. Therefore, these high concentrations could have been caused by meteorological build-up, and it is possible the model set-up underestimated the strength of this effect.

The exceptional concentrations measured especially at Harwell (and to a lesser extent at Detling) could have been caused by (i) missing local sources in the area, (ii) over-reporting of the concentrations by AMS measurements or by the PMF analysis applied to apportion measured OA into its components, (iii) meteorological build-up, or (iv) a combination of these. However, the specific origin of the large discrepancy between model and measurements at the Harwell site during these four days remains unknown.

### 3.4 Hourly evaluation statistics during the rest of the ClearfLo winter IOP, 2012

Table 4 presents the hourly evaluation statistics at the Detling, Harwell, and London North Kensington sites during the winter IOP (as in Fig. 8) but excluding the period of largely unexplained high SFOA concentrations episode between 13 Jan–18 Jan. These *r*-values (0.35–0.53; range of hourly *r*-values for all three sites) are lower than the daily-average *r*-values in comparison





with the annual datasets shown in Figs. 5 and 6 (0.59–0.78). This is expected as the diurnal emissions profile used for all European countries assigns equal amounts of residential combustion emissions to the morning and evening which does not match the measured diurnal SFOA profiles at these three sites (as was shown in Fig. 7).

For Detling, the *Base* case run underestimates the mean measured SFOA concentration by −27% (NMB), the *Base4x* experiment results in an overestimation of +49%, and the *combRedist* yields a close match with −3%. For Harwell, the *Base* scenario has a NMB of −36%, which becomes an overestimation of +64% with the *Base4x* experiment. The *combRedist* emissions have a minor effect on modelled concentrations at Harwell (NMB of −31%; compared with −36% for the *Base* case; the other evaluation statistics of *combRedist* are also close to those of the *Base* case). At North Kensington, hourly comparison shows similar results for the two experiments as was also seen in the daily evaluation. Both emissions cases capture the mean concentration well, but the *Base4x* experiment yields a better *r*-value than *combRedist* experiment .

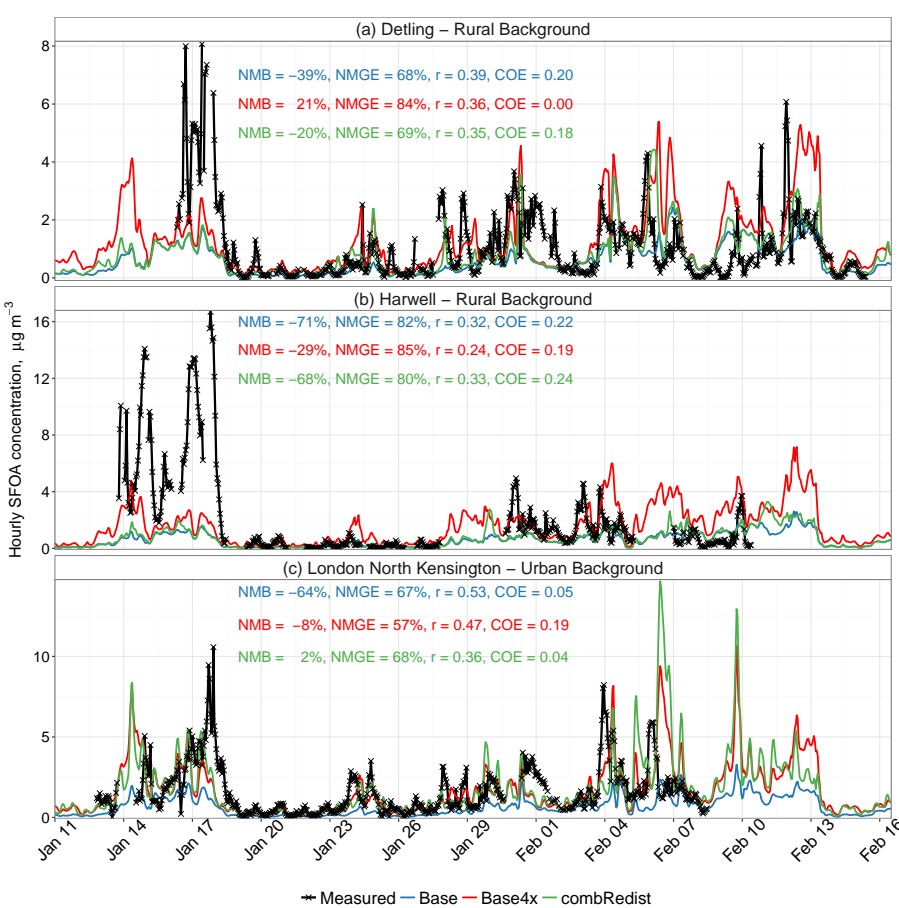

**Figure 8.** Time-series of measured and modelled hourly average SFOA concentrations at the (a) Detling, (b) Harwell, and (c) London North Kensington measurement sites, winter IOP 2012. Note the different scales on the y-axis.





**Table 4.** Evaluation statistics for modelled vs measured hourly-average concentrations of SFOA during the ClearfLo winter IOP measurement sites as in Fig. 8, but excluding the period of 13 Jan–18 Jan.

| Site | Experiment | NMB | NMGE | $r$ | COE |
|---|---|---|---|---|---|
| Detling | *Base* | −27% | 66% | 0.38 | 0.19 |
| | *Base4x* | 48% | 92% | 0.45 | −0.14 |
| | *combRedist* | −3% | 68% | 0.41 | 0.16 |
| Harwell | *Base* | −36% | 69% | 0.43 | 0.18 |
| | *Base4x* | 64% | 105% | 0.42 | −0.24 |
| | *combRedist* | −31% | 68% | 0.44 | 0.19 |
| North Kensington | *Base* | −64% | 66% | 0.53 | 0.06 |
| | *Base4x* | −1% | 56% | 0.53 | 0.20 |
| | *combRedist* | 12% | 73% | 0.35 | −0.04 |

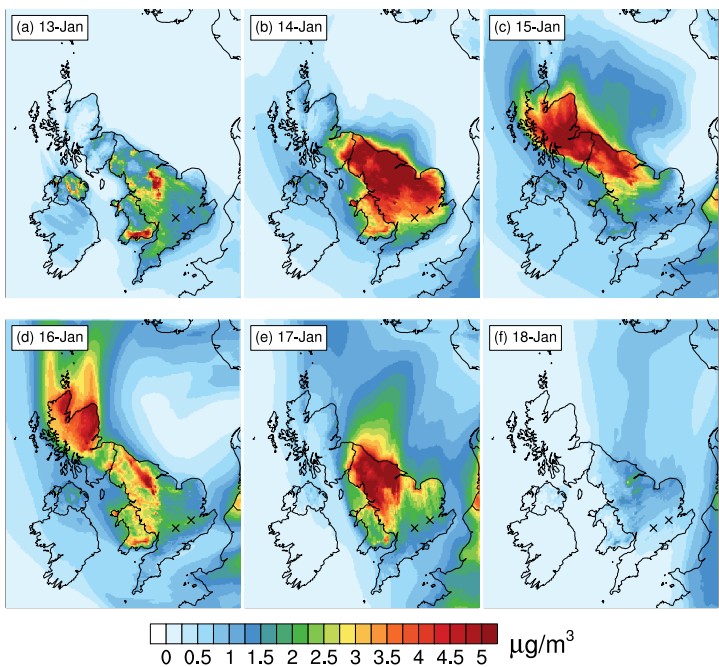

**Figure 9.** Daily-average modelled (*Base4x* experiment) SFOA surface concentrations during the episode of high SFOA concentrations at the beginning of the winter IOP, year 2012. The black crosses mark the measurement site locations, left: Harwell, right: London North Kensington.





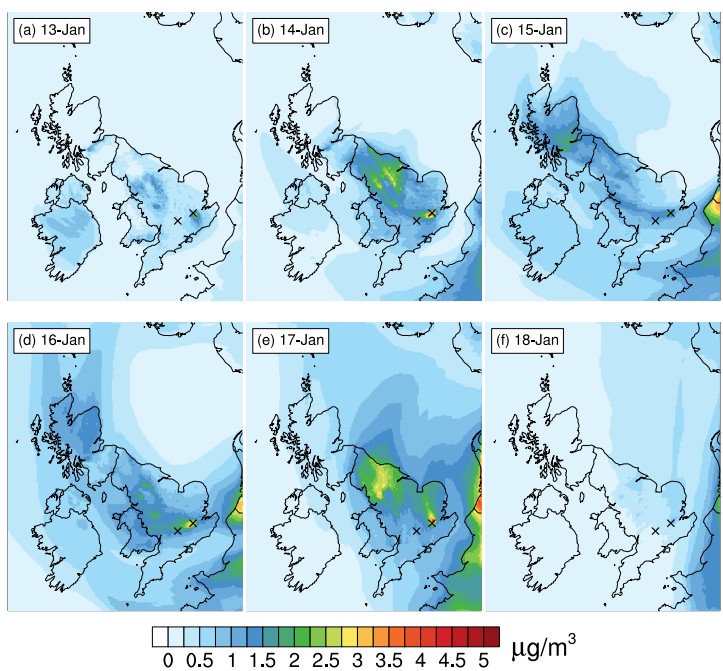

**Figure 10.** Same as Fig. 9, but for the *combRedist* experiment.

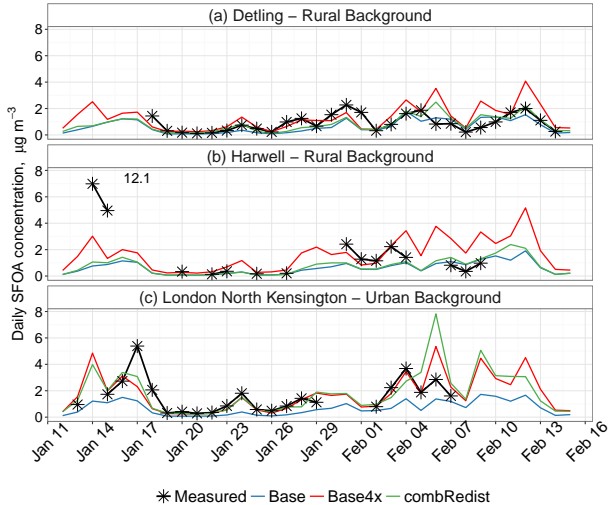

**Figure 11.** Daily-average measured and modelled SFOA concentrations at the (a) Detling, (b) Harwell, and (c) North Kensington measurement sites, year 2012. On panel (b), the concentration measured at 16-Jan is given as a text label.



### 3.5 Comparison of modelled EC with measured EC and BC

Locations for the UK black carbon (BC) measurement network existing in 2012 are shown in Fig. 12. All of the measurement sites use an Aethalometer to measure optically-absorbing aerosol on a filter tape. Optical absorption is converted to an effective black carbon (eBC, referred to as just BC in this work) concentration using a mass-specific absorption cross-section (MAC).

Three sites (Harwell, London North Kensington, and London Marylebone Road) also have daily-average filter measurements that are analysed for elemental carbon (EC) with a thermal optical technique in which the split between EC and OC can vary depending on instrumental and analysis parameters. Comparisons of the daily measurements of EC-R (measured EC corrected using reflectance), EC-T (corrected using transmittance), and daily-averaged BC are presented in Appendix A, along with a discussion of the sources of method biases. Substantial discrepancies in reported concentrations exist (at times, more than a

factor of 2). Furthermore, the sign and the magnitude of discrepancies differ by season and by measurement site. Consequently, detailed (i.e. hourly or daily) model-measurement evaluation is not justified, and only seasonally-averaged concentrations are presented in this section.

Seasonal boxplots of daily-average concentrations of measured BC, EC-T, EC-R, and modelled concentrations of EC for Harwell, North Kensington, and Marylebone Road are shown in Fig. 13. In addition to emissions from domestic heating, these

concentrations also include all other sources of EC, mainly traffic. Within each panel, the different datasets were made to be of equal size, i.e. days with missing measurements, or with measurements below the limit of detection for EC-R or EC-T values, were also removed from measured BC as well as from the modelled time series. Although the pollutant levels measured close to the traffic source at the Marylebone Road roadside site and the modelled concentrations are not fairly comparable due to the differences in spatial scale (major road vs 5 km model resolution), they are included to illustrate the range of concentrations in

a megacity.

For Harwell, modelled EC concentrations from the *Base* and *combRedist* experiments are similar, but concentrations from the *Base4x* experiment are higher. This is consistent with the comparison of modelled and measured SFOA concentrations in the *Base4x* experiment which overestimates wood and coal burning contributions at the two rural sites near London (Harwell and Detling, excluding 13–18-Jan). Fig. 13 shows that the *Base* and *combRedist* modelled EC concentrations are close to

measured EC-R during all seasons except autumn for which overestimations of several pollutants for many different days and periods are discussed in Ots et al. (2016a). Generally, measured BC concentrations are higher than measured or modelled EC, but during winter, measured BC, measured EC-R and modelled EC (*Base* or *combRedist* experiments) are a close match, whereas measured EC-T is substantially lower.

At North Kensington, both the *Base4x* and *combRedist* experiments result in similarly higher concentrations than the *Base*

run, and these experiments match the measurements of EC-R well. At Marylebone Road, modelled concentrations are substantially lower than are measured. This is expected as Marylebone Road is a roadside site with heavy traffic flows and the concentrations captured at the measurement site are therefore much higher than a 5 km × 5 km × 40 m model grid cell average.

Seasonal boxplots of daily-average concentrations of measured BC and modelled EC at all the other BC network measurement sites are presented in Fig. 14. For these urban sites, both the *combRedist* case and the *Base4x* case yield a higher modelled





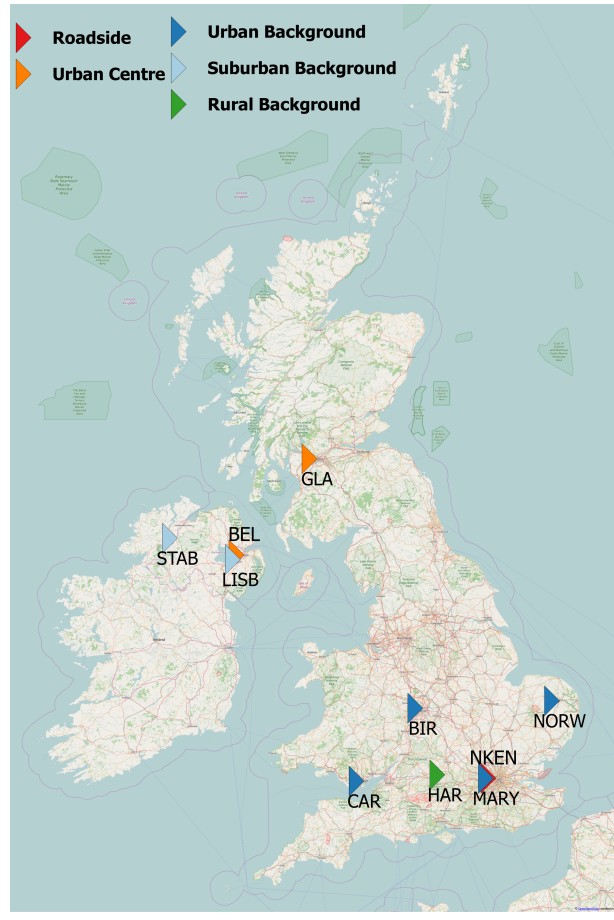

**Figure 12.** BC-network measurement site locations in 2012. Site names are abbreviated as follows: GLA - Glasgow Centre, STAB - Strabane, BEL - Belfast Centre, LISB - Lisburn Dunmurry, BIR - Birmingham Tyburn, NORW - Norwich Lakenfields, CAR - Cardiff, HAR - Harwell, NKEN - London North Kensington, MARY - London Marylebone Road. Underlying map from © OpenStreetMap contributors.

concentration than the *Base* run and bring the modelled EC into better agreement with measured BC. There are no sites for which the *combRedist* yields a lower modelled concentration than the *Base* run (i.e. for these, urban, sites, the redistribution does not make the comparison worse). Therefore, based on BC measurements at these sites in different parts of the UK (which are on average higher than measurements of EC-T or EC-R), the experiments conducted for the investigation of spatial distri-
5    bution of residential wood and coal burning do not result in unrealistic EC concentrations. The three sites in Northern Ireland (BEL, LISB, and STAB in Fig. 14) exhibit a stronger seasonal cycle in the measurements than the other sites (i.e. relatively greater increase for autumn and winter), indicating stronger traditions of residential solid fuel burning in this part of the UK.





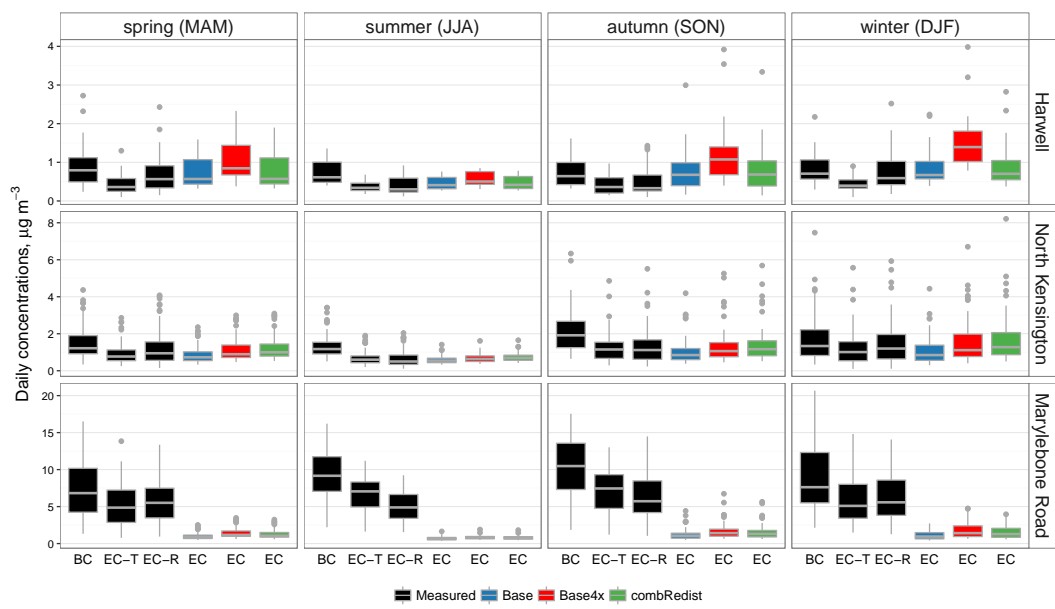

**Figure 13.** Seasonal boxplots of daily-average concentrations of measured BC, EC-T, EC-R, and modelled EC concentrations at Harwell, London North Kensington, and London Marylebone Road, the three measurement sites that measure both BC and EC, year 2012. Middle line: median, boxes: 25th and 75th quartiles (i.e. the interquartile range - IQR), whiskers extend to 1.5×IQR, and all daily values beyond the whiskers are plotted individually.



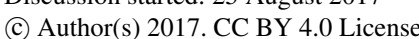


**Figure 14.** Seasonal boxplots of daily-average concentrations of measured BC and modelled EC concentrations at the BC-network measurement sites, year 2012. Site name abbreviations are given in Fig. 12. Middle line: median, boxes: 25th and 75th quartiles (i.e. the interquartile range - IQR), whiskers extend to 1.5×IQR, and all daily values beyond the whiskers are plotted individually.



## 4   Conclusions

In this study, different assumptions for the spatial distribution and total emitted amount of PM emissions from solid fuel burning in the UK were tested with the EMEP4UK atmospheric chemical transport model. These model experiments were conducted to investigate the large model underestimations of SFOA concentrations compared with aerosol source apportion-
ment measurements which arise when using the officially-reported PM emissions inventory (a NMB of $-71\%$ at the London North Kensington urban background site, for example). The two main scenarios considered were *Base4x*, and *combRedist*. For *Base4x*, officially reported $PM_{2.5}$ from the SNAP2 emission source sector (residential and other non-industrial combustion) were increased by a factor of 4. For the *combRedist* experiment, half of the emissions from SNAP2 were redistributed linearly by residential population density to extend emissions into smoke control areas. The emission total for the *combRe-*
*dist* experiment was the same as that officially reported (i.e. equal to the *Base* scenario, and 4 times less than in the *Base4x* scenario).

Comparison of model output with AMS-PMF measurements of SFOA concentrations at an urban background and roadside site in central London (a smoke control area), shows that *Base4x* yielded better daily and hourly correlations than the *combRedist*. Therefore, for certain air masses, the spatial distribution reported by the national emissions inventory appears reasonable.
However, the *Base4x* overestimated SFOA concentrations at the rural sites, whereas the *combRedist* better captured mean measured concentrations across the range of site types and locations. The *combRedist* was intentionally simplistic (exactly 50% of the national total was spatially redistributed), so a better agreement might be, for example, *Base2x* + 30% redistributed to population density (or another combination of *Base* emissions and redistribution to include emissions for smoke control areas). The results also suggest that refinement of the prescribed temporal profiles for residential emissions may also be re-
quired as the measurements indicated higher levels of SFOA concentrations in the evening than in the morning, whereas the emissions profiles used here emitted relatively more during the morning. It is acknowledged that the experiments undertaken here investigated only potential discrepancies in the national atmospheric emissions inventory, and not other potential sources of model-measurement discrepancy such as shortcomings in model quantification of dry and wet deposition, both of which also depend on accurate model description of the meteorology.
Modelled concentrations of elemental carbon were compared with different measurements: EC-R, EC-T, and black carbon (BC). Substantial discrepancies were noted between the different measurements of this component of PM, so detailed comparison with the model was not presented. However, based on seasonal-average concentrations at the BC network sites over the UK, it was shown that the concentrations derived from the *combRedist* experiments improved the comparisons of modelled vs observed concentrations. Therefore, some redistribution of SNAP2 emissions into smoke control areas, as also suggested by
the SFOA comparisons, appears justified. Overall, our results suggest that simulations of SFOA can be improved by adjusting the spatial distribution of the national emissions inventory.





## 5 Code availability

The EMEP model is OpenSource and can be downloaded from www.emep.int.

## 6 Data availability

Processed measurement data used in this study are available through the ClearfLo project archive at the British Atmospheric
Data Centre (http://badc.nerc.ac.uk/browse/badc/clearflo). The model data (input, code, relevant output) are archived at the
University of Edinburgh and are available on request.

## Appendix A: Measured EC-R, EC-T, BC

### A1 Overview

Because of its diverse origins and chemical processing, airborne carbonaceous particulate matter exists in a continuum of differ-
ent forms – from pure graphitic-like elemental carbon (EC) at one end through to an array of highly chemically-functionalised
organic compounds at the other (Gelencsér, 2004). This raises the issue of at what level of chemical oxidation/functionalisation
should EC no longer be categorised as EC but as organic carbon (OC). In practice, EC, and an alternative measure for the 'sooty
carbon' content of particulate matter – black carbon (BC) – are both measurement defined, rather than chemically defined (Pet-
zold et al., 2013; Lack et al., 2014). EC is defined by thermal methods (heating up a PM sample and burning off the more
volatile organic components such that what is only burnt off at highest temperatures is called EC; Chow et al. (2007)), and BC
is defined by optical methods (measuring how opaque the material is to transmission of visible or near infrared wavelengths to
determine effective black carbon, or eBC; Bond and Bergstrom (2006)). So by definition EC and eBC are not measuring the
same thing, and both methods require choices in the quantification process. For example, the EC method requires choice on
the temperature programme and pyrolysis correction approach used to assign a distinction between EC and OC (Chow et al.,
2004; Cavalli et al., 2010), while the eBC method requires a 'shadowing' correction (Virkkula et al., 2007) and imposition
of a mass-specific absorption coefficient (MAC) to convert optical absorbance to mass concentration (Andreae and Gelencsér,
2006; Quincey et al., 2009). These issues have been the subject of much discussion in the literature, some of which is cited
above, with development of standard protocols and terminology. In general, however, measurements of EC and eBC on the
same samples are very highly correlated, and to a first approximation EC and eBC data values can both be used in comparison
of model output of EC against measurements. For the remainder of this Appendix, eBC is refered to as BC.

### A2 Measurements of EC-T and EC-R

The UK Particle Numbers and Concentration Network collects daily samples of $PM_{10}$ onto binder-free pure quartz filters
using a Partisol 2025 sampler at three measurement stations: London Marylebone Road (kerbside), London North-Kensington
(urban background) and Harwell (rural background). These filters are analysed at the UK National Physics Laboratory (on a




Sunset Laboratory Carbon Aerosol Analysis Lab instrument). The protocol used to quantify total carbon (TC) of the sample is a variation of the NIOSH protocol known as Quartz (Quincey et al., 2009). During the heating of the sample, some organic matter will be converted to elemental carbon by pyrolysis. This conversion is monitored by continuously measuring the transmittance (T) and reflectance (R) of the sample. The T or R signal is used to apportion TC into OC and EC by taking account of

carbonaceous material that was OC, but become pyrolised into EC during the heating process. However, the quantification of the pyrolysed OC differs whether T or R is used, adding uncertainty to the measurements.

Figure A1 illustrates the uncertainty of the EC split from TC using transmittance (EC-T) and EC split from TC using reflectance (EC-R). For Harwell and North Kensington, EC-R is higher than EC-T (on average over the full dataset) which agrees with the findings of Chow et al. (2004). However, it can be noted that for both Harwell and North Kensington, in the

lower range ($<$ ~0.4 µg m$^{-3}$ for Harwell, $<$ ~0.6 µg m$^{-3}$ for North Kensington), EC-T is higher than EC-R, whereas above these values, EC-R is higher than EC-T.

Figures A2, A3, and A4 split the points in Fig. A1 into seasons for Harwell, North Kensington, and Marylebone Road, respectively. In spring and winter, EC-T is lower than EC-R (NMB from $-7\%$ to $-35\%$), whereas during summer EC-T is higher than EC-R (NMB from $+19\%$ to $+31\%$). In autumn, average measured concentrations of EC-R and EC-T are similar

(NMB from $-2\%$ to $+10\%$), but substantial scatter can be seen for some days as measured concentrations of EC-T and EC-R can differ from each other by more than a factor of 2. These consistent findings at all three sites suggest that the different methods for quantifying of EC with the same instrument are dependent on season.

## A3   Measurements of BC

Measurements of BC use an Aethalometer (Magee AE22) and involve the collection of $PM_{2.5}$ onto a quartz tape, for which

the absorption $\alpha$ [m$^{-1}$] is measured by single-pass attenuation of 880 nm light, corrected for scattering. The absorption is converted to BC concentration using a mass-specific absorption cross section (MAC). The measurements are reported with an hourly timestep. The assumption of a single value for the MAC can lead to uncertainties due to (Heal and Quincey, 2012):

– Atmospheric oxidation changing the MAC of the sample (making it appear less dark).

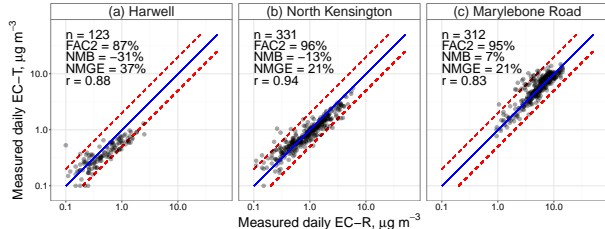

**Figure A1.** Scatterplot of measured daily-average EC split from TC using transmittance (EC-T) and EC split from TC using reflectance (EC-R), year 2012. Data below the detection limit ($<$ 0.1 µg m$^{-3}$) have been removed, leading to the lower number of data points for Harwell.



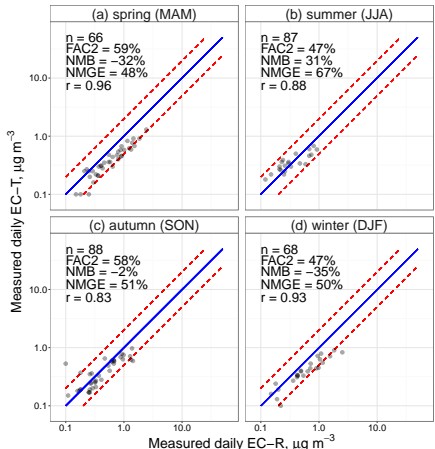

**Figure A2.** Scatterplot of measured daily-average EC split from TC using transmittance (EC-T) and EC split from TC using reflectance (EC-R) at the Harwell site split by seasons, year 2012.

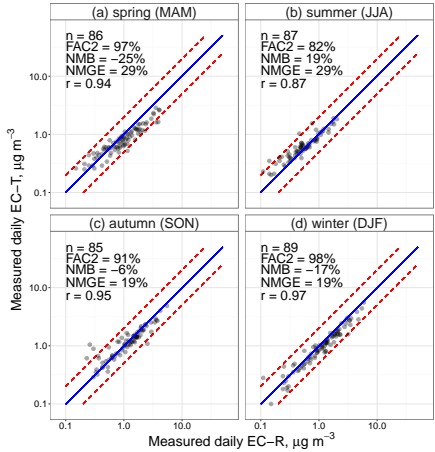

**Figure A3.** Scatterplot of measured daily-average EC split from TC using reflectance (EC-R) against EC split from TC using transmittance (EC-T) at the North Kensington site split by seasons, year 2012.

– Variations in the size distribution of the particles leading to variations in the MAC.

– Variations in the external and internal mixing with other particles in the sample leading to variations in the optical properties of the sample.

## A4   Measured BC vs measured EC

5  Figure A5 shows daily-average time-series of measured EC-R and BC for the three measurement sites that have both sets of measurements. For Jan–March and Oct–Dec, measurements of BC and EC-R at the Harwell and North Kensington sites are



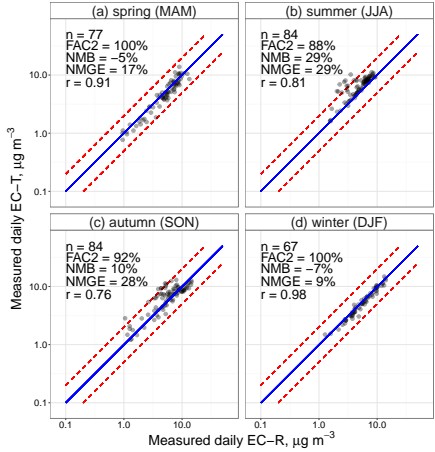

**Figure A4.** Scatterplot of measured daily-average EC split from TC using reflectance (EC-R) against EC split from TC using transmittance (EC-T) at the Marylebone Road site split by seasons, year 2012.

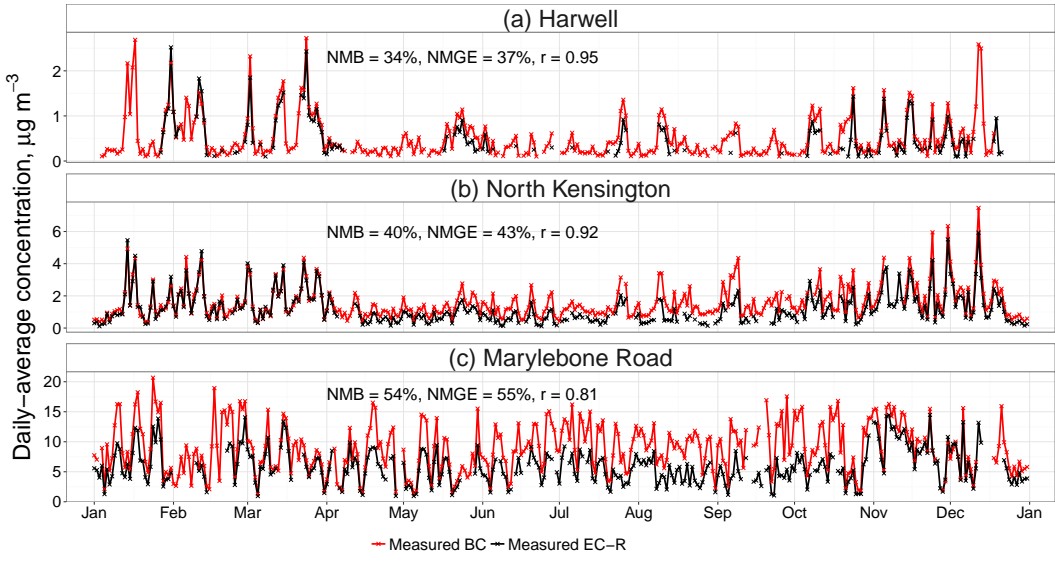

**Figure A5.** Time-series of daily-average measured EC-R and BC concentrations at the (a) Harwell, (b) North Kensington, and (c) Marylebone Road measurement sites, year 2012.

a close match to each other, whereas from April to September, BC is consistently higher than EC-R. At Marylebone Road, winter and early spring months have better agreement than the summer, but BC is overestimated compared to EC-R throughout the year.




## A5 Appendix summary

There are several inherent and methodological uncertainties in quantifying the refractory part of carbonaceous aerosol as EC or BC. Overall, the mean absolute values of the different measurements relate to each other as follows: BC > EC-R > EC-T, but the magnitudes of the differences, and in some cases also the order, vary for seasons and for individual sites.

*Author contributions.* R.O. designed, executed and analysed the model experiments. R.O. and M.R.H. wrote the manuscript. L.X., L.R.W., S.C.H., and N.L.N. collected and processed the AMS measurements at the Detling Winter IOP site, D.E.Y., J.D.A, H.C, and D.C.G. at the London North Kensington site, and E.M., C.M, and A.D. at the London Marylebone Road site and the Harwell site. M.V manages the EMEP4UK project. S.R. processed the population density data. I.A.M. processed the Base case emissions inventory. All authors contributed to the final version of the manuscript.

*Competing interests.* The authors declare that they have no conflicts of interest.

*Disclaimer.* None.

*Acknowledgements.* The authors acknowledge the UK Department for Environment, Food and Rural Affairs (Defra) and the Devolved Administrations for funding aspects of the development of the EMEP4UK model (AQ0727), for partial support for the aerosol measurements, as well as access to the AURN data, which were obtained from uk-air.defra.gov.uk and are subject to Crown 2014 copyright, Defra, li-
cenced under the Open Government Licence (OGL). Partial support for the EMEP4UK modelling from the European Commission FP7 ECLAIRE project is gratefully acknowledged. This work was supported in part by the UK Natural Environment Research Council (NERC) ClearfLo project [grant ref. NE/H003169/1]. R.O. was supported by a PhD studentship (University of Edinburgh and NERC-CEH contract 587/NEC03805). D.E.Y. was supported by a NERC PhD studentship [ref. NE/I528142/1]. L.X., L.R.W., S.C.H., and N.L.N. acknowledge support from US Department of Energy (grant no. DE-SC000602).
NCAR command language (NCL) was used to produce the maps (NCAR, 2015), and R, openair and ggplot2 for the analysis and all other plots (R Core Team, 2016; Carslaw and Ropkins, 2012; Wickham, 2009).





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
