# Peer review of "Modelling carbonaceous aerosol from residential solid fuel burning with different assumptions for emissions"

_Atmospheric Chemistry and Physics, 2017_

## Referee Comment (RC1) · Anonymous Referee #1 · 22 Sep 2017

This is a very useful contribution to an ongoing and important debate on the impacts on PM concentrations of wood burning.

Abstract: It doesn't mention the Redist analysis. Wouldn't it be worth saying that a simple redistribution of emissions according to population is not correct?

Comment on whether the degree day factors actually reflect the use of these wood burners would be useful.

Whilst there is an under prediction at K&C and the Waters paper (3x the emissions) - most experiments are using base emissions of NAEI, why is this? I have read the Waters paper and it not only gives the 3x factor but also the wood use in different

[Figure]

UK regions. You did not use these data but could you comment on how the different scenarios you did run compare?

P13 line 3-4 - What about European assumptions bearing in mind the results in Belgium. Could you comment on how important the long range transport of these emissions are and could be if the results in Belgium and UK are reflected more widely in Europe?

It would be good to provide a quick comparison of the Marylebone and Kensington site results. Looking at the map they are close to each other and I guess they have very similar SFOA concentrations. Is this the case? In addition, whilst I realize that you have used what data is available for the UK, could you say something about the limitations in addressing the wood burning emissions inventories UK wide using a small number of sites close to each other in the SE of England.

Page 11 line 9 - where it says for more discussions see Ots 2016a, why not just add a sentence discussing the measurement uncertainty?

Page 11 fig 7 - it is clear that the diurnal profile of SFOA is similar at all sites and not reflected in the currently used emissions profile for this source. I have read the work of Fuller in London which showed there to be a strong evening peak in emissions from domestic wood burning, especially at weekends. It would have been good to test an alternative emissions profile, which better reflects the burning of wood and would have been helpful for other model users.

Fig 8. results - 14th-15th Jan was also a weekend could you comment on the likely weekday to weekend use of wood burners as well as the weekend evening use. What was the evening temperature during these events?

Fig 11 - Comment on Detling daily data. The daily average measurements around the 17th Jan doesn't seem to be reflected in the plot. Is this because of <75% data capture?

Figure 13. I don't really see much point in comparing the results of the model at Marylebone Road, so you should remove this plot.

Figure 14. Could you rescale these plots? You can barely make out the modelled EC in many of them?

[Figure]

---

## Referee Comment (RC2) · Anonymous Referee #2 · 29 Sep 2017

The paper contains interesting information on important sources of emissions in a major urban area. I have one major concern, but after attention to the points below this paper should be suitable for publication on ACP.

My main worry concerns the assumption that the SFOA (and other POA) emissions are inert. In most VBS modelling studies such emissions are allocated to a number of VBS bins, and allowed to evaporate and react with OH. Further, the results presented for London in Xu et al. (2016) do not show any large SFOA contribution to the low-volatility OM mass, suggesting that the high fractions found in Young et al were of semi-volatile OA. Assuming inert emissions will tend to overestimate the PM concentrations
associated with this POA. The authors should re-visit and investigate the implications of their inert assumption.

Connected to this, what is the likely status of the emission measurements behind the SFOA inventory for the UK? Do the techniques used to produce the emission factors include condensables? With so much focus on one emission category, and the fact that condensables are a 'hot' topic (Denier van der Gon et al, 2015, Ciarelli et al., 2017), the authors should inform the readers more about such properties.

This issue of volatility and associated uncertainties seems to be ignored throughout the manuscript.

Other comments

Page 2, L10. The Bergstrom reference is a PhD Thesis. Give the published papers instead.

Page 2, L5-15. What about emissions from cooking?

Page 2, L16. What is the 'Great London Smog' - give a reference.

Page 2, L33. I believe Belgium has also included condensables in their emissions estimates, which brings me back to the point raised above.

Page 3, Sect. 2.1: The text should give some details about the SOA framework used here. What assumptions are made about SVOC, IVOC, and aging? What was done for ASOA and BSOA?

Page 3. The statistics given for model performance are useful, but they seem only to refer to London. How about elsewhere, since this paper deals with the UK as a whole?

Page 4, Add the ion labels for SO4, NH4 and NO3. (For example NO3 is a gaseous compounds important for night-time chemistry, whereas I think the authors mean the nitrate ion.)

Page 5, L4 claims that Ots et al. (2016a) showed that modelled SFOA were substantially underestimated at North Kensington, but according to Table 3 of that paper the SFOA PMF factors were convolved with the OOA2 factors.

Page 6, Fig. 2. Units of Mg/km2 would be easier for comparison with other studies.

Page 11, Sect 3.2. Measured profiles of SFOA result from a mixture of emissions profiles, atmospheric dispersion, and PMF interpretation. The model should capture the first two, but I wonder how much PMF contributes. For example, if the diurnal emissions profile is responsible for the concentrations profile, why would SFOA emissions peak around midnight for N. Kensington?

Page 12, L13 and associated text and Figures. Were these "exceptional" concentrations also seen for other pollutants, for example NO2. Would model performance for other components help the analysis here? (Also, the word exceptional seems a bit excessive here. Are such concentrations really so infrequent?)

Page 21. The WRF model is also open source, and details should be included here. I think section 5 and 6 could also be merged, since the code is mentioned in both. Currently it is confusing though, since Sect. 5 says code should be obtained from www.emep.int, but Sect. 6 says code is from the University of Edinburgh.

Appendices: This type of information is typically provided as Supplementary material.

---

## Referee Comment (RC3) · Anonymous Referee #3 · 19 Oct 2017

The manuscript by Ots et al. presents a modeling study that explores the uncertainty of the residential and non-industrial combustion emissions sector over the UK and Ireland. The uncertainty estimates, translated into sensitivity experiments in the manuscript, are driven from past studies (Ots et al., 2016) and comparisons with measurements. The model domain covers the whole UK and Ireland, although the analysis is heavily based on comparisons with data in London.

The work presented is a standard modeling approach, where an emissions sector is perturbed and the different model versions are compared against measurements, to evaluate which of the scenarios under study is performing best against some met-

ric, which in this case it is a fraction of organic aerosols (solid fuel OA; SFOA) and black/elemental carbon. The analysis does not have any mistakes, although one can argue that the approach of non-volatility and ageing for SFOA deserves improvement, especially given the temperature-dependent parameterization presented for the emissions during cold days. The results are not surprising either; a low bias in SFOA is improved by increasing its emissions, and rather linearly, as seen in Table 3. For both sites studied, the NMB presented in Table 3 for the combRedist experiment is roughly equal to the arithmetic mean of the Base and Redist simulations, which is what one would expect from the experimental design. In addition, no attempts have been made to link this work with either the study in Belgium mentioned in the manuscript, or with other relevant areas, limiting the scope of the work as presented. The last sentence of the conclusions also supports my concern about the limited scope of the study, again, as presented in the manuscript. Regardless, I believe that the work is sound and deserves publication in ACP after addressing my comments below.

Specific comments

p. 3, l. 25: The FINN inventory has natural fires only, or all open burning? Some of the open biomass burning is anthropogenic (e.g. deforestation, agricultural fires).

Section 2.2: The way I understand the degree-day factors equation is that it does not affect days with temperature higher than 18 C, but it increases emissions for colder days. If this is correct, isn't it going to increase the annual totals? In addition, why not apply the same approach hourly (section 3.2) and get a more natural diurnal variability, instead of the imposed one?

p. 5, l. 17: Please start a new paragraph with "The experiments Base, . . ."

Section 3.1: How far apart are the two stations? Are they in adjacent gridboxes, the same one, really far away? How about differences in local influences, if any?

p. 11, l. 16-17: Why only correlation and not the other metrics? More generally, this

is an important piece of information and should be expanded, even though it is already published.

Figure 7: Some error bars or other means that present temporal variability can be very informative here.

The AMS instruments mentioned also measure total PM1 OA. It would have been very informative if the discussion included a comparison with those data as well, either (preferably) alongside the comparisons with SFOA, or (at least) in the same way the BC/EC comparison is presented.

p. 16, l. 18-19 and Figure 13, last row: This site does not add anything to the discussion, I recommend to remove it.

p. 20, l. 28: I am not entirely convinced that "the combRedist experiments improved the comparisons". Only the negative NMB was really targeted with the experimental design, and it is expected that increased emissions of an inert aerosol tracer will increase aerosol levels at surface, especially close to sources, thus reducing (or even eliminating) the negative bias. Figures 13 and 14, which represent a more regional picture, do not show any significant improvement for that particular simulation either.

Appendix A1 contains textbook information and it is not necessary, although it consists of a nice collection of references and the discussion is fluid, so I am hesitating to propose to remove it. Appendices A2-A4 should be supplementary material. Appendix A5 should move in section 3.5.

Technical corrections

p. 5, l. 4: Is Hjj correct (so please explain) or it should have been Hdd?

p. 9, l. 9: Please take this sentence out of the parentheses.
* * *

---

## Author Comment (AC1) · 1 Dec 2017

We thank the reviewer for their very supportive comments. We respond to each comment individually below. The reviewer's comments are in italics and blue font, our responses are in normal text.

*This is a very useful contribution to an ongoing and important debate on the impacts on PM concentrations of wood burning.*

1. *Abstract: It doesn't mention the Redist analysis. Wouldn't it be worth saying that a simple redistribution of emissions according to population is not correct?*

We have included the following in the Abstract:
"A third experiment, *Redist* - all emissions redistributed linearly to population density - is also presented as an indicator of the maximum concentrations an assumption like this could yield."
And the following in the Conclusions:
"A third experiment, *Redist* - all emissions redistributed linearly to population density – is also presented as an indicator of the maximum concentrations an assumption like this could yield. It is recognised however that this is not realistic as the most densely populated areas (of large apartment buildings) are unlikely to have many individual fireplaces."

2. *Comment on whether the degree day factors actually reflect the use of these wood burners would be useful.*

Yes, people in London do use wood and coal for heating purposes. We have now included the relevant references in "Section 2.2: Model experiments":
"Recent studies in London have shown that local contributions of SFOA coincide with days of low temperature (Fuller et al., 2014; Crilley et al., 2015). Therefore, degree-day factors were included to modulate the daily variation in emissions from the SNAP2 sector according to ambient temperature (i.e. increasing the emissions during colder days)."

3. *Whilst there is an under prediction at K&C and the Waters paper (3x the emissions) - most experiments are using base emissions of NAEI, why is this? I have read the Waters paper and it not only gives the 3x factor but also the wood use in different UK regions. You did not use these data but could you comment on how the different scenarios you did run compare?*

The Waters' paper is certainly very relevant and should be used to inform a re-evaluation of the National Atmospheric Emissions Inventory (NAEI) assumptions about the extent and spatial distribution of wood and coal burning in urban and rural areas over the UK. However, the paper is a summary of survey information on domestic combustion habits not an emissions database. To derive emissions of SFOA would require combining data and assumptions about amount of wood burned, the appliance types and a range of emission factors, which is well beyond the objectives of our paper. Secondly, the data in the Waters paper were presented for 12 regions (e.g. London, East Midlands, Scotland, Wales, etc.) rather than the much finer grid that could be used in a national atmospheric chemical transport model (e.g. we aggregate NAEI's 1kmx1km emissions to our model grid – 5kmx5km). Again,

the production of new gridded SFOA emissions is well beyond the objectives of our paper, but we note in conclusion of our work that the Waters paper and our modelling study help inform development of updated national emissions inventories for domestic combustion PM.

4. *P13 line 3-4 - What about European assumptions bearing in mind the results in Belgium. Could you comment on how important the long range transport of these emissions are and could be if the results in Belgium and UK are reflected more widely in Europe?*

We have added the following sentences to the top of "3 Results and Discussion" (as that is where we first mentioned European import):
"In our experiments, we did not modify European emissions – we used exactly what has been reported to the CEIP. While there is reason to believe European emissions of SFOA are also underreported, we do not believe this to have a major influence on the surface concentrations of SFOA over the UK as even our Base4x experiment (Fig. 3b) only indicates very modest regional transport of our SFOA to Europe.

5. *It would be good to provide a quick comparison of the Marylebone and Kensington site results. Looking at the map they are close to each other and I guess they have very similar SFOA concentrations. Is this the case? In addition, whilst I realize that you have used what data is available for the UK, could you say something about the limitations in addressing the wood burning emissions inventories UK wide using a small number of sites close to each other in the SE of England.*

This is a good idea and we have now included the annual average measured concentrations in the captions of Figures 5 and 6: "The annual average measured SFOA concentrations at these sites were 0.9 µg m$^{-3}$ at Marylebone Road, and 1.0 µg m$^{-3}$ at North Kensington." We agree that only using sites from the same area (Southern England) is not ideal but these datasets are really rather unique – especially in its length as well as the fact that for January 2012 we have 4 sites operating simultaneously, two of each type (urban and rural). However, acknowledging the limitations of using a small number of sites close to each other is exactly why we later present comparisons with the Aethalometer data, including sites from a national network.

6. *Page 11 line 9 - where it says for more discussions see Ots 2016a, why not just add a sentence discussing the measurement uncertainty?*

Agreed, we have added the following sentences:
"For example, Ots et al. 2016a presented scatter plots of daily averaged concentrations of the different OA components derived from measurements with the two different AMS instruments at the North Kensington site during the winter IOP (the cToF-AMS versus the HR-ToF-AMS). While these comparisons showed good correlations between the two measurements (0.88 to 0.95 for the primary OA components, 0.77 for secondary OA), on some days the absolute measured concentrations of specific components do differ, sometimes by more than a factor of two."

7. *Page 11 fig 7 - it is clear that the diurnal profile of SFOA is similar at all sites and not reflected in the currently used emissions profile for this source. I have read the work of Fuller in London which showed there to be a strong evening peak in emissions from domestic wood burning, especially at weekends. It would have been good to test an alternative emissions profile, which better reflects the burning of wood and would have been helpful for other model users.*

We agree that the diurnal cycles specified for these emissions need to be modified. We did something very similar in our cooking OA paper (Ots et al. 2016b); cooking oil/meat frying OA is a source that is currently not included in European emissions at all. However, for the work in hand, we felt that including experiments for the spatial distribution and total amount was already quite a lot to present, and that the overall -71% normalised mean bias of modelled vs measured SFOA at London North Kensington should be addressed before the more finer temporal-scale issue of hourly emissions profiles. We acknowledge and call for further work on this in several places in the manuscript, including in the Abstract and the Conclusions.

8. *Fig 8. results - 14th-15th Jan was also a weekend could you comment on the likely weekday to weekend use of wood burners as well as the weekend evening use. What was the evening temperature during these events?*

This is a very good point to make, thank you!
We have included a note about this (plus our subsequent observation that ambient temperatures were somewhat lower during this period) in the Results (italics was already there, **bold font text we have added now**):
"*Nevertheless, southern England did experience a sustained high-pressure weather system during these days,* **including noticeably lower temperatures for 14-18 Jan than average (Crilley et al. 2015: Fig.2).** *Sustained high pressure usually leads to a very stable atmosphere with descending air masses. Therefore, these high concentrations could have been caused by meteorological build-up, and it is possible the model set-up underestimated the strength of this effect.* **Furthermore, 14-15 Jan was a weekend whereby people are more likely to spend time home and therefore potentially use their fireplaces more than on weekdays."**

9. *Fig 11 - Comment on Detling daily data. The daily average measurements around the 17th Jan doesn't seem to be reflected in the plot. Is this because of <75% data capture?*

Yes that is the reason. We have now expanded the information about the threshold where Fig. 11 is first introduced:
"Time-series of daily-average concentrations during the winter IOP are shown in Fig. 11. At Detling, the measurements commenced the morning of 16-Jan (Fig. 8a) but since we used a data capture threshold of 75%, 16-Jan and 17-Jan did not include sufficient hourly data points to present measured daily averages for this site."

10. *Figure 13. I don't really see much point in comparing the results of the model at Marylebone Road, so you should remove this plot.*

The paper provides the following explanation for presenting the data shown in Figure 13.
"Although the pollutant levels measured close to the traffic source at the Marylebone Road

roadside site and the modelled concentrations are not fairly comparable due to the differences in spatial scale (major road vs 5 km model resolution), they are included to illustrate the range of concentrations in a megacity."

We believe that the above remains a relevant reason for presenting the comparisons at Marylebone Road also.

*11. Figure 14. Could you rescale these plots? You can barely make out the modelled EC in many of them?*

We have now added some vertical space to these plots, as well as given the borders of the boxes (previously grey) the same colour as the insides so the very thin ones become more noticeable. The final ACP paper has more space per page than the ACPD format (55 lines vs 35 lines) so the editorial staff will have more space to stretch the figure as well (our image format is PDF so typesetting it on a larger surface will not lose quality). The description of Figure 14 now says "Each box is the interquartile range - IQR," rather than what is on Figure 13: "*Middle line: median, boxes: 25th and 75th quartiles (i.e. the interquartile range - IQR),*" as you could not make out the median line on most of the very thin modelled boxes anyway.

---

## Author Comment (AC2) · 1 Dec 2017

We thank the reviewer for their supportive comments on the interest and suitability of the work for publication in ACP. We respond to each comment individually below. The reviewer's comments are in italics and blue font, our responses are in normal text.

*The paper contains interesting information on important sources of emissions in a major urban area. I have one major concern, but after attention to the points below this paper should be suitable for publication on ACP.*

1. *My main worry concerns the assumption that the SFOA (and other POA) emissions are inert. In most VBS modelling studies such emissions are allocated to a number of VBS bins, and allowed to evaporate and react with OH. Further, the results presented for London in Xu et al. (2016) do not show any large SFOA contribution to the low- volatility OM mass, suggesting that the high fractions found in Young et al were of semi- volatile OA. Assuming inert emissions will tend to overestimate the PM concentrations associated with this POA. The authors should re-visit and investigate the implications of their inert assumption.*

The reason this study does not present model simulations with the volatile treatment of SFOA is that for the AMS-PMF data, primary (SF)OA and oxygenated (secondary) OA are separated. Therefore, the direct comparisons with SFOA measurements here do not include the semivolatile components as those would only become condensed after atmospheric ageing but then they would be measured as oxygenated OA, not SFOA. The volatile components and secondary OA precursors are not needed to test the main hypothesis of this paper – that the spatial distribution of wood and coal burning emissions should not be assumed to be zero in smoke control areas. Using primary components to demonstrate this is sufficient.

This is not to say, of course, that SFOA emissions do not include precursors for SOA. The inclusion of semivolatile SOA precursors from SFOA is of course necessary to close the gap between total measured OA and total modelled OA. Indeed, the work by Xu et al. (2016) to which the reviewer refers above (and other work) acknowledge that oxygenated OA likely contains secondary and/or aged SFOA.

However, having reviewed our original text towards the end of Section 2.1 ("In all the experiments presented here, SFOA is assumed to be non-volatile and it does not undergo atmospheric ageing") we acknowledge that we were not clear on the point we make above. We have therefore added another sentence so that this description now reads: "In all the experiments presented here, SFOA is assumed to be non-volatile and does not undergo atmospheric ageing. This is not because it is assumed that there is no aging of SFOA emissions but because the model simulations compare against AMS-PMF determinations of primary SFOA concentrations."

2. *Connected to this, what is the likely status of the emission measurements behind the SFOA inventory for the UK? Do the techniques used to produce the emission factors include condensables? With so much focus on one emission category, and the fact that condensables are a 'hot' topic (Denier van der Gon et al, 2015, Ciarelli et al., 2017), the authors should inform the readers more about such properties.*

*This issue of volatility and associated uncertainties seems to be ignored throughout the manuscript.*

We have indicated in our response to comment 1 that we have not 'ignored' the issue of volatility because it is not important but because it is not needed in this examination of spatial patterns and magnitudes of primary SFOA $PM_{2.5}$ emissions against primary SFOA measurements. We agree that the issue of 'condensables' in measuring and reporting PM levels is indeed a 'hot' topic. This issue applies to both ambient measurements and to the emissions at source that underpin emissions factors and inventories. It also an issue that is not just confined to solid-fuel burning. For example, one of our previous papers examined the influence of IVOC emissions associated with diesel vehicles (determined from ambient measurements but not currently included in emissions inventories) on the generation of additional anthropogenic SOA: Ots et al. (2016) "Simulating secondary organic aerosol from missing diesel-related intermediate-volatility organic compound emissions during the Clean Air for London (ClearfLo) campaign", Atmos. Chem. Phys., 16, 6453-6473. The measurement and modelling communities have long been aware of a need to attempt harmonisation in measurement conditions and reporting of PM and associated condensable emissions (e.g. dilution tunnel conditions etc.) across source sectors and across countries. The issue is raised in various fora, e.g. CEIP and TFMM, but it is challenging and slow to agree and implement. In this study we included the best-available county-by-country domestic sector $PM_{2.5}$ emissions as reported to CEIP. This includes instances of countries that do report condensables as part of their PM emissions, e.g. Belgium (who report these as part of the total PM emissions, not separately), and countries that don't. Without considerable more coordination of measurement and reporting it not possible to make any evidence-based adjustments to the CEIP emissions. In the revised paper we have added a further sentence towards the end of Section 2.1 (after the new sentence inserted in response to comment 1): "It is likely that domestic $PM_{2.5}$ emissions reported to CEIP vary by country according to whether condensables are included in the $PM_{2.5}$ emissions but the specific information is not known."

*Other comments*
3. *Page 2, L10. The Bergstrom reference is a PhD Thesis. Give the published papers instead.*

Thank you for pointing this out. We now only include the following relevant published reference from the thesis: Denier van der Gon et al. 2015.

4. *Page 2, L5-15. What about emissions from cooking?*

We believe it extremely unlikely that there are significant primary PM emissions associated with use of solid fuels solely for the purpose of domestic cooking in the UK and the rest of Europe. (We assume that this what the reviewer intended by their question.) We have not encountered any literature to suggest that solid fuel for cooking is a source that needs to be considered. This is not to deny that some domestic solid-fuel heat sources may also be used for cooking. It is certainly the case, however, that there are PM emissions associated with the cooking process, but these have different chemical signature and are categorised separately by AMS-PMF measurements. We have previously undertaken model investigations of this cooking source: Ots et al. (2016) "Model simulations of cooking organic aerosol (COA) over

the UK using estimates of emissions based on measurements at two sites in London", Atmos. Chem. Phys., 16, 13773-13789.

5. *Page 2, L16. What is the 'Great London Smog' - give a reference.*

We have now included the following reference:
Bell, M. L., Davis, D. L., and Fletcher, T.: A retrospective assessment of mortality from the London smog episode of 1952: the role of influenza and pollution, Environmental Health Perspectives, 112, 6–8, http://www.ncbi.nlm.nih.gov/pmc/articles/PMC1241789/, 2004.

6. *Page 2, L33. I believe Belgium has also included condensables in their emissions estimates, which brings me back to the point raised above.*

Yes, Belgium does include condensables in their total PM emissions estimates from this source, but not separated out. We have responded in detail to the earlier version of this comment above (comment no. 2).

7. *Page 3, Sect. 2.1: The text should give some details about the SOA framework used here. What assumptions are made about SVOC, IVOC, and aging? What was done for ASOA and BSOA?*

The modelling framework for ASOA and BSOA is as described in the literature cited in the paper, viz:
Ots, R. et al.: Simulating secondary organic aerosol from missing diesel-related intermediate-volatility organic compound emissions during the Clean Air for London (ClearfLo) campaign, Atmos. Chem. Phys., 16, 6453-6473, 2016.
Simpson, D. et al.: The EMEP MSC-W chemical transport model - technical description, Atmos. Chem. Phys., 12, 7825-7865, 2012.
As we respond above in response to comment no. 1, we do not undertake any 'aging' treatment for SNAP 2 emissions because the AMS-PMF data provide separate data for primary (SF)OA and oxygenated (secondary) OA. Condensable components of SNAP2 emissions are therefore not needed to test the main hypothesis of this paper – that the spatial distribution of wood and coal burning emissions should not be assumed to be zero in smoke control areas. Using primary components to demonstrate this is sufficient. In order to make this point more directly in our revised paper we have added text to Section 2.1, as we have detailed in our responses above.

8. *Page 3. The statistics given for model performance are useful, but they seem only to refer to London. How about elsewhere, since this paper deals with the UK as a whole?*

We agree that only using sites from the same area (Southern England) is not ideal but these datasets are really rather unique – especially in their length, time resolution and chemical speciation, as well as the fact that for January 2012 we have 4 sites operating simultaneously, two of each type (urban and rural). However, acknowledging the limitations of using a small number of sites close to each other is exactly why we later present comparisons with the Aethalometer data, including sites from a national network.

9. *Page 4, Add the ion labels for SO4, NH4 and NO3. (For example NO3 is a gaseous compounds important for night-time chemistry, whereas I think the authors mean the nitrate ion.)*

Thank you for pointing out this error. The ions are now labelled.

10. *Page 5, L4 claims that Ots et al. (2016a) showed that modelled SFOA were substantially underestimated at North Kensington, but according to Table 3 of that paper the SFOA PMF factors were convolved with the OOA2 factors.*

The data reported in Table 3 was from the following paper, where the measurement dataset and the derivation of the different factors is described in detail: Young, D. E., Allan, J. D., Williams, P. I., Green, D. C., Flynn, M. J., Harrison, R. M., Yin, J., Gallagher, M. W., and Coe, H.: Investigating the annual behaviour of submicron secondary inorganic and organic aerosols in London, Atmos. Chem. Phys., 15, 6351–6366, doi:10.5194/acp-15-6351-2015, 2015.

In the study by Young et al. (2015), a year of measurements were performed using the c-ToF-AMS (compact) and it is the PMF factors from that dataset that contain the two convolved PMF factors. Section 4.4 of the paper describes how these two convolved factors were dealt with. In brief, both factors had a strong and similar diurnal cycle, so the effect of being convolved was reduced by using daily averaged concentrations. In the current work presented here, these daily average concentrations are used for the annual comparisons, thus the issue of the convolved factors should not be significantly influencing the overall observations. In contrast, concentration data from the HR-ToF-AMS (high-resolution) instrument are used for the hourly comparisons presented in the current work. The PMF factors resulting from the HR-AMS data were not found to be convolved, which is likely due to a combination of the fact that the measurements were high resolution and the HR-ToF-AMS was deployed only during the intensive observation periods, rather than the full year like the c-ToF-AMS. Consequently, hourly comparisons presented in this work use the HR-ToF-AMS data.

In summary, whilst we acknowledge the general issue of convolved factors, it is not likely influencing the overall results in the current study since datasets of the appropriate time resolution were used for each of the cases and each of these datasets had been treated so as to reduce the issue.

11. *Page 6, Fig. 2. Units of Mg/km2 would be easier for comparison with other studies.*

We have added the following to the caption: "20 Mg per (our) grid square is 0.8 Mg km$^{-2}$"

12. *Page 11, Sect 3.2. Measured profiles of SFOA result from a mixture of emissions profiles, atmospheric dispersion, and PMF interpretation. The model should capture the first two, but I wonder how much PMF contributes. For example, if the diurnal emissions profile is responsible for the concentrations profile, why would SFOA emissions peak around midnight for N. Kensington?*

We anticipate that the peak around this time is due to the product of the impacts of greater evening local emissions and reducing boundary layer height and increasing atmospheric stability during night-time.

13. *Page 12, L13 and associated text and Figures. Were these "exceptional" concentrations also seen for other pollutants, for example NO2. Would model performance for other components help the analysis here? (Also, the word exceptional seems a bit excessive here. Are such concentrations really so infrequent?)*

These concentrations (e.g. 12 $\mu$g m$^{-3}$ daily average at Harwell) are indeed exceptional as this is just one component. The recommended maximum daily average concentration for total PM$_{2.5}$ is no more than 10 $\mu$g m$^{-3}$ and that must accommodate all components: organic and inorganic, not just SFOA. Furthermore, our Figures 5 shows that daily average concentrations throughout the year exceed 4 $\mu$g m$^{-3}$ only a few days a year.

Comparisons with other pollutants are presented in Figure 6 in:
Ots, R., et al.: Simulating secondary organic aerosol from missing diesel-related intermediate-volatility organic compound emissions during the Clean Air for London (ClearfLo) campaign, Atmos. Chem. Phys., 16, 6453–6473, doi:10.5194/acp-16-6453-2016, 2016a.

14. *Page 21. The WRF model is also open source, and details should be included here. I think section 5 and 6 could also be merged, since the code is mentioned in both. Currently it is confusing though, since Sect. 5 says code should be obtained from www.emep.int, but Sect. 6 says code is from the University of Edinburgh.*

The ACP submission instructions request these two separate sections. Where we mention "code" in Section 6, we do not mean the core source code but the scripts used to initialise a model simulation (i.e. linking all our input files to the model executable). We have now changed the word "code" to "scripts" in Section 6.

15. *Appendices: This type of information is typically provided as Supplementary material.*

We will take the advice of the ACP editor (or production team) on this point.

---

## Author Comment (AC3) · 1 Dec 2017

We thank the reviewer for their support that our work is sound and deserving of publication in ACP after response to their comments. We respond to each comment individually below. The reviewer's comments are in italics and blue font, our responses are in normal text.

*The manuscript by Ots et al. presents a modeling study that explores the uncertainty of the residential and non-industrial combustion emissions sector over the UK and Ireland. The uncertainty estimates, translated into sensitivity experiments in the manuscript, are driven from past studies (Ots et al., 2016) and comparisons with measurements. The model domain covers the whole UK and Ireland, although the analysis is heavily based on comparisons with data in London.*

*The work presented is a standard modeling approach, where an emissions sector is perturbed and the different model versions are compared against measurements, to evaluate which of the scenarios under study is performing best against some metric, which in this case it is a fraction of organic aerosols (solid fuel OA; SFOA) and black/elemental carbon. The analysis does not have any mistakes, although one can argue that the approach of non-volatility and ageing for SFOA deserves improvement, especially given the temperature-dependent parameterization presented for the emissions during cold days. The results are not surprising either; a low bias in SFOA is improved by increasing its emissions, and rather linearly, as seen in Table 3. For both sites studied, the NMB presented in Table 3 for the combRedist experiment is roughly equal to the arithmetic mean of the Base and Redist simulations, which is what one would expect from the experimental design. In addition, no attempts have been made to link this work with either the study in Belgium mentioned in the manuscript, or with other relevant areas, limiting the scope of the work as presented. The last sentence of the conclusions also supports my concern about the limited scope of the study, again, as presented in the manuscript. Regardless, I believe that the work is sound and de- serves publication in ACP after addressing my comments below.*

*Specific comments*

1. *p. 3, l. 25: The FINN inventory has natural fires only, or all open burning? Some of the open biomass burning is anthropogenic (e.g. deforestation, agricultural fires).*

Yes, thank you. We have now changed "natural fires" to "open burning (including wild fires and agricultural burning)".

2. *Section 2.2: The way I understand the degree-day factors equation is that it does not affect days with temperature higher than 18 C, but it increases emissions for colder days. If this is correct, isn't it going to increase the annual totals? In addition, why not apply the same approach hourly (section 3.2) and get a more natural diurnal variability, instead of the imposed one?*

Yes, the reviewer's understanding is correct, it does not increase the annual total. Regarding the second question (why not apply this on hourly temperatures):

Firstly, the work we present in this manuscript did not create and make methodological changes to how the degree-day factors, we used them as described in Simpson et al. (2012): The EMEP MSC-W chemical transport model - technical description, *Atmos. Chem. Phys.*, 12, 7825–7865, doi:10.5194/acp-12-7825-2012.

Secondly, while we agree that the hourly profiles used in this study (also based on the above reference) need review (in our abstract we write: "The model results also suggest the assumed temporal profiles for residential emissions may require review to place greater emphasis on evening (including 'discretionary') solid-fuel burning.") we do not think hourly variation in temperature has a very direct effect on increased heating. This may be true for thermostat-based central heating systems (gas or electricity `fuelled`), but solid fuel burning means that people need to be home and awake (i.e. even if the temperature drops overnight, people will not wake up and start burning wood and coal). Therefore, the hourly profile of SFOA emissions is unlikely to be driven by changes in temperature on an hourly basis.

*3.   p. 5, l. 7: Please start a new paragraph with "The experiments Base, . . ."*

Agreed and done.

*4.   Section 3.1: How far apart are the two stations? Are they in adjacent gridboxes, the same one, really far away? How about differences in local influences, if any?*

The two stations are about 4.5 km apart, in separate adjacent model grid cells. Detailed maps of these sites (overlaid with residential and workday population density at 1 km resolution) can be found in Fig. 1 in Ots et al. (2016) "Model simulations of cooking organic aerosol (COA) over the UK using estimates of emissions based on measurements at two sites in London", Atmos. Chem. Phys., 16, 13773-13789, reproduced below:

[Figure]

**Figure 1.** Residential **(a)** and workday **(b)** population density in central London at 1 km × 1 km resolution. The residential population maps are based on Reis et al. (2016). While the same methodology is applied to derive workday population maps, they are not yet published due to delays in the provision of workday population census data for Scotland. Also shown are the measurement sites and the EMEP4UK 5 km × 5 km grid used in this study (white lines). Underlying map contains Ordinance Survey (OS) data © Crown Copyright 2015.

We have now included the following in Section 3.1 (paragraph 2) of this manuscript:
"These two sites are ~4.5 km apart, in adjacent model grid cells, and represent different kinds of urban areas. North Kensington is comparatively residential whereas Marylebone Road is near central London and therefore exhibits very high numbers of people during the workday. More discussion on this, including detailed maps can be found in Ots et al. (2016b)."

*5.   p. 11, l. 16-17: Why only correlation and not the other metrics? More generally, this is an important piece of information and should be expanded, even though it is already published.*

Agreed, we have added the following sentences:

"For example, Ots et al. 2016a presented scatter plots of daily averaged concentrations of the different OA components derived from measurements with the two different AMS instruments at the North Kensington site during the winter IOP (the cToF-AMS versus the HR-ToF-AMS). While these comparisons showed good correlations between the two measurements (0.88 to 0.95 for the primary OA components, 0.77 for secondary OA), on some days the absolute measured concentrations of specific components differ, sometimes by more than a factor of two."

6. *Figure 7: Some error bars or other means that present temporal variability can be very informative here.*

Overall, we agree with this comment that it is useful to try to present the variability, and not just the final aggregated (in this case the mean) values. However, since we plot 4 individual lines on each panel here, we think that adding for example the standard deviation around each line would make the information on the panels overlap each other and the plot would become `too busy`. A solution could be to separate pairs of these lines on separate plots (e.g. as we've done in Figs. 5 and 6) but since the focus of this paper is not diurnal profiles we feel that increasing the figure count and the total length of this paper for this is not justified.

7. *The AMS instruments mentioned also measure total PM1 OA. It would have been very informative if the discussion included a comparison with those data as well, either (preferably) alongside the comparisons with SFOA, or (at least) in the same way the BC/EC comparison is presented.*

Comparisons with other components of OA (HOA and ASOA) are presented already in our previous publication: Ots, R., et al.: Simulating secondary organic aerosol from missing diesel-related intermediate-volatility organic compound emissions during the Clean Air for London (ClearfLo) campaign, Atmos. Chem. Phys., 16, 6453–6473, doi:10.5194/acp-16-6453-2016, 2016a. The AMS instruments also assign concentrations to a COA (cooking OA) component which are not routinely modelled because there are no cooking emissions in the emissions inventory for the model. However, in another paper we used the AMS-measured COA concentrations for a model exploration of potential magnitude and spatial distribution of primary COA emissions: Ots et al. "Model simulations of cooking organic aerosol (COA) over the UK using estimates of emissions based on measurements at two sites in London", Atmos. Chem. Phys., 16, 13773-13789, 2016b.

8. *p. 16, l. 18-19 and Figure 13, last row: This site does not add anything to the discussion, I recommend to remove it.*

The paper provides the following explanation for presenting the data shown in Figure 13. "Although the pollutant levels measured close to the traffic source at the Marylebone Road roadside site and the modelled concentrations are not fairly comparable due to the differences in spatial scale (major road vs 5 km model resolution), they are included to illustrate the range of concentrations in a megacity."

We believe that the above remains a relevant reason for presenting the comparisons at Marylebone Road also.

> *9. p. 20, l. 28: I am not entirely convinced that "the combRedist experiments improved the comparisons". Only the negative NMB was really targeted with the experimental design, and it is expected that increased emissions of an inert aerosol tracer will increase aerosol levels at surface, especially close to sources, thus reducing (or even eliminating) the negative bias. Figures 13 and 14, which represent a more regional picture, do not show any significant improvement for that particular simulation either.*

We agree that the Base4x experiment only targets the negative model NMB, for the reason the reviewer states: "*it is expected that increased emissions of an inert aerosol tracer will increase aerosol levels at surface*". Hence the aim of the combRedist experiment: namely, to investigate the effect of redistribution of some of the SFOA emissions on both the negative NMB (caused by the fact that the NAEI does not currently assign emissions to smoke control areas), as well as on the other mod-obs metrics presented in this manuscript (r, NMGE, COE), i.e. the redistribution of SFOA sources affects SFOA concentrations in the different air masses arriving at these sites, not just the emissions assigned to the local grid cell. For example, in Detling and Harwell, combRedist had a much lower NMGE and a much better COE than Base4x (Table 4). Furthermore, Figures 13 and 14 show that these tests do not cause completely unnecessary effects in, for example, Northern Ireland, i.e., that we are not completely removing SFOA from lower populated areas and simply ending up with it in the most densely populated areas of London.

> *10. Appendix A1 contains textbook information and it is not necessary, although it consists of a nice collection of references and the discussion is fluid, so I am hesitating to propose to remove it. Appendices A2-A4 should be supplementary material. Appendix A5 should move in section 3.5.*

We remain of the opinion that the material in this section contains relevant information to the interpretation of our comparisons of modelled EC with measured BC, the latter of which is measured at many more sites geographically spread across the UK, than with measurements of EC which are made only at 3 sites (2 of which are in London). We believe it is important to challenge the model with the greater amount of measured BC data but also wish to remind readers of issues associated with equating BC and EC. We will take the advice of the ACP editor (or production team) on whether this material is most appropriate as an Appendix or as Supplementary Material, but given its relative brevity we would prefer to keep it as an Appendix.

*Technical corrections*
> *11. p. 5, l. 4: Is Hjj correct (so please explain) or it should have been Hdd? p. 9, l. 9: Please take this sentence out of the parentheses.*

Yes, it should have been Hdd, not Hjj. We have now fixed this, thank you!

---

## Author Response (AR2)

**acp-2017-568: Modelling carbonaceous aerosol from residential solid fuel burning with different assumptions for emissions**
**Ots, R. et al.**

We thank the reviewer for their further review of our paper and are happy to provide more information in our responses here and to the manuscript. The most substantial clarifying additions can be found in Section 2.1 – Model description (including emissions description) and Section 2.3 – Comparison with measurements. The reviewer's comments are presented in full below, in blue font.

**Response to further review comments**

*The authors have provided lengthy replies, but I am afraid I still have two main issues with this paper:*
*i) I do not believe the authors have adequately addressed my comments about volatility. As explained below, they ignore the possibly very significant effect of evaporation of SFOA emissions. I would like to know what the UK POA emissions are, and to what extent they contain condensables. This is a reasonable request I think for a paper that focuses on such emissions from one country.*

The UK POA emissions do not include condensables; we apologise for not making this explicit in our responses to the previous version in which we had argued that model simulations with (semi)volatile SFOA emissions were not required for our model-measurement comparison. We have now added text to Section 2.1 Model Description as follows.

First, we have changed the sentence that originally read "*It is likely that domestic PM$_{2.5}$ emissions reported to CEIP vary by country according to whether condensables are included in the PM$_{2.5}$ emissions but the specific information is not known*" to now read "*The UK emissions inventory for domestic PM$_{2.5}$ does not include condensables, but this information is not known for the emissions reported to CEIP by other countries.*"

Secondly, we have now added the following paragraph to this section:
"*The reason this study does not present model simulations with the volatile treatment of SFOA is that for the AMS-PMF data, primary (SF)OA and oxygenated (secondary) OA are separated. Therefore, the direct comparisons with SFOA measurements here do not include the semivolatile components as those would only become condensed after atmospheric ageing but then they would be measured as oxygenated OA, not SFOA. The volatile components and secondary OA precursors are not needed to test the main hypothesis of this paper – that the spatial distribution of wood and coal burning emissions should not be assumed to be zero in smoke control areas. Using primary components to demonstrate this is sufficient. This is not to say that SFOA emissions do not include precursors for SOA. The inclusion of semivolatile SOA precursors from SFOA is of course necessary to close the gap between total measured OA and total modelled OA. Indeed, the work by Xu et al. (2016) acknowledge that oxygenated OA likely contains secondary and/or aged SFOA.*"

*ii) The authors are responding to the referees, but often without any change in the paper (most replies to my comments were to me, not readers). The usual procedure is to change the paper*

*to clarify the points raised by the referees, since one cannot expect readers to churn through the whole ACPD discussion.*

We had made additions to the paper where we thought these necessary but are happy to include more information in the manuscript (detailed below) whilst remaining conscious not to overly lengthen a paper that is already quite long. We have also referenced relevant previous studies as sources of further information for readers.

Our responses to the further specific comments made in this second review are given below. First, however, we indicate where we have added further material to the paper relating to comments made at the first review.

In Section 2.1, we have now directly referred to the published descriptions of the modelling framework for ASOA and BSOA by inserting the following sentence "*Descriptions of the modelling framework for ASOA and BSOA are given in Simpson et al. (2012) and Ots et al. (2016).*" before the sentence beginning "*In all the experiments presented here…*"

In Section 3.2, where text originally read "*This temporal misclassification of emissions for the UK is expected to have a detrimental effect on all of the model hourly evaluation statistics, except NMB*" we have now expanded this to read "*Whilst reducing boundary layer height is likely also to contribute to greater measured concentrations in late evening, the temporal misclassification of emissions for the UK is expected to have a detrimental effect on all of the model hourly evaluation statistics, except NMB.*"

Regarding the question in the previous review about convolved factors, we have now included the following text to Section 2.3 after this sentence that is currently in the paper: Limitations and uncertainties of these measurement datasets have been discussed in Ots et al. (2016a, b).
"*In addition, the original analysis of the annual c-ToF-AMS (compact) dataset in Young et al. (2015a) identified that the SFOA and OOA factors were convolved. Section 4.4 of Young et al. (2015a) describes how these two convolved factors were dealt with. In brief, both factors had a strong and similar diurnal cycle, so the effect of being convolved was reduced by using daily averaged concentrations. In the current work presented here, these daily average concentrations are used for the annual comparisons, thus the issue of the convolved factors should not be significantly influencing the overall observations. In contrast, concentration data from the HR-ToF-AMS (high-resolution) instrument are used for the hourly comparisons presented in the current work. The PMF factors resulting from the HR-AMS data were not found to be convolved, which is likely due to a combination of the fact that the measurements were high resolution and the HR-ToF-AMS was deployed only during the intensive observation periods, rather than the full year like the c-ToF-AMS. Consequently, hourly comparisons presented in this work use the HR-ToF-AMS data.*"

*In more depth on the first point, the authors comment that:*

*a) ..we have not 'ignored' the issue of volatility because it is not important but because it is not needed in this examination of spatial patterns and magnitudes of primary SFOA PM2.5 emissions against primary SFOA measurements.*
*b) In this study we included the best-available county-by-country domestic sector PM2.5 emissions as reported to CEIP. This includes instances of countries that do report condensables as part of their PM emissions, e.g. Belgium (who report these as part of the total*

*PM emissions, not separately), and countries that don't. Without considerable more coordination of measurement and reporting it not possible to make any evidence-based adjustments to the CEIP emissions.*
*While the statement in (a) about spatial patterns may be true, the one about magnitudes certainly is not. It would only be true if the emissions included in PM2.5 (especially in the UK) were indeed non-condensable. This may be true for some countries, and is not for others, but this paper does not demonstrate the UK situation in any way. The statement in (b) does not help, and the comments about European emissions are misleading I think. This study is not really about European emissions; the focus is very much on the UK and indeed London. With such a focus I would have expected a proper explanation of what the United Kingdom does with regard to reporting condensables or not in its PM2.5 inventory. Did the authors speak to the emission inventory experts in the UK to clarify this problem?*

We respond to all these comments together. (i) Yes we did speak to a UK national emissions expert and confirmed that the UK PM$_{2.5}$ emissions do not include condensables. As indicated above, we have now added this fact explicitly to the paper. In addition, one of the authors of this paper, Jeroen Kuenen, is an emissions expert. We prefer not to refer to unpublished work too much which is why we used Belgium as an example of a country that does include condensables in their PM emissions reported to CEIP, and which is published.

(ii) As stated on the first page of our responses, we have now added a paragraph to 'Section 2.1 Methods Description' to explain why this study does not consider condensable emissions, the paragraph starting: *"The reason this study does not present model simulations with the volatile treatment of SFOA…."*

(iii) We have also now added the following extra text about emissions to Section 2.1 Model Description:
*"Furthermore, the various sampling methods used to derive emission factors (which are applied by each country reporting emissions to CEIP) vary greatly (Denier van der Gon et al., 2015). The two main types are filter measurements (capturing only solid particles), and dilution tunnel measurements (capturing solid particles and condensable organics). The difference between the two methods can be large - up to 5-fold for wood-burning (Denier van der Gon et al., 2015), which is similar to what was shown by May et al. (2013) that up to 80% of the mass of POA from biomass burning may evaporate when diluted from plume to ambient conditions. Based on partitioning calculations for a range of OA (from 0.01 to $10^5$ ug m$^{-3}$) in Donahue et al. (2006), only the 10 ug m$^{-3}$ volatility bin exhibits a substantial portions in both gaseous and solid form. The organic components in all other bins are either mostly solid or mostly gaseous. May et al. (2013) also shows that only 10% of biomass burning emissions (Table 2 in May et al. (2013)) is in the 10 ug m$^{-3}$ bin. Therefore, the potential overestimation arising from not letting these solids in the UK emissions inventory evaporate is no more than 10% (as the lower volatility materials is almost completely solid, and the higher volatility material is almost completely gaseous)."*

(iv) In the previous revision we had also added text to make the point that without considerable more coordination of measurement and reporting (of domestic PM$_{2.5}$ emissions) it is not possible to make any evidence-based adjustments to the CEIP emissions.

*It is not clear that the authors have understood the implications of condesable/semi-volatile*

*POA concepts, since they only discuss (mainy in the reply to referees) aging. Evaporation is very likely te main issue with this work, not aging. According to May et al 2013, between 50-80% of the mass of POA from biomass burning may evaporate when diluted from plume to ambient conditions. The AMS instruments measure ambient air, so any study that ignores this will necessarily give too big a contribution from SFOA.*

The measured amounts of SFOA against which we compare our model simulations are taken directly from the AMS-PMF speciation of OA components into HOA, SFOA, OA, COA. The AMS instrument measures only the condensed (aerosol particle) material in the ambient air. In the *Base* run, the model underestimates the measured (condensed) SFOA concentrations by 71% on average. In our simulation experiments we show various ways (by increasing the total amount as well as redistributing) to bring the model values of SFOA closer to what the AMS quantifies for this component. The non-volatile treatment of SFOA in the model does not make the AMS see more SFOA than there really is. If a proportion of the emissions going into the model are in reality being lost through evaporation then there would be greater discrepancy between model and measurement to account for.

In summary: the evaporation of semivolatile SFOA does not affect how much condensed SFOA the AMS measures. Our work is exploring ways of closing a model underestimation in condensed SFOA, not an overestimation.

*Finally, I also saw the other referees suggestion to remove the Marleybone road plot from Fig.13, and agree with this. Having stated that 'the modelled concentrations are not fairly comparable due to differences in spatial scale (major road vs 5 km model resolution)', there is no valid argument to compare I think. The authors suggest that one needs to illustrate the range of concentrations in a megacity, but plenty of other studies do that. And major roads have little to do with SFOA modelling.*

Response: We have now removed Marylebone Road from Figure 13 and deleted the text in this section that referred to that particular dataset.

[revised manuscript text omitted]

---

## Author Response (AR3)

**acp-2017-568: Modelling carbonaceous aerosol from residential solid fuel burning with different assumptions for emissions**
**Ots, R. et al.**

**Response to additional review comments**

*The revised manuscript is much improved, and it is especially good to know the status of the UK's OA emissions inventory, and that condensables are not included.*

Response: Thank you!

*I just saw one issue with the revised text. In Sect. 2.1, the additional text which starts 'Based on partitioning calculations for a range of OA ....' should be deleted, since the assumptions about a 10 ug/m3 bin are not relevant and this estimation therefore incorrect. The concentrations relevant to a consideration of a 'May et al' type evaporation of POA are those of the dilution chamber, which are far higher than 10 ug/m3.*

Response: We have removed this text.

*As the authors now state clearly that they are dealing with inert SFOA emissions, I suggest they delete this 10 ug/m3 calculation, and re-arrange the text a little. I would have started with comments about the potential importance of evaporation (citing May et al, Deniier van der Gon et al) but then gone on to explain that these complications are not needed since the UK has just inert SFOA in its inventory.*

Response: We have removed the calculation and re-arranged the text as suggested (e.g. the UK SFOA sentence now follows the May and van der Gon references).

Descriptions of the modelling framework for ASOA and BSOA are given in Simpson et al. (2012) and Ots et al. (2016a). In all the experiments presented here, SFOA is assumed to be non-volatile and it does not undergo atmospheric ageing.

The reason this study does not present model simulations with the volatile treatment of SFOA is that for the AMS-PMF data, primary (SF)OA and oxygenated (secondary) OA are separated. Therefore, the direct comparisons with SFOA measurements here do not include the semivolatile components as those would only become condensed after atmospheric ageing but then they would be measured as oxygenated OA, not SFOA. The volatile components and secondary OA precursors are not needed to test the main hypothesis of this paper – that the spatial distribution of wood and coal burning emissions should not be assumed to be zero in smoke control areas. Using primary components to demonstrate this is sufficient. This is not to say that SFOA emissions do not include precursors for SOA. The inclusion of semivolatile SOA precursors from SFOA is of course necessary to close the gap between total measured OA and total modelled OA. Indeed, the work by Xu et al. (2016) acknowledge that oxygenated OA likely contains secondary and/or aged SFOA.

Furthermore, the various sampling methods used to derive emission factors (which are applied by each country reporting emissions to CEIP) vary greatly (Denier van der Gon et al., 2015). The two main types are filter measurements (capturing only solid particles), and dilution tunnel measurements (capturing solid particles and condensable organics). The difference between the two methods can be large - up to 5-fold for woodburning (Denier van der Gon et al., 2015), which is similar to what was shown by May et al. (2013) that up to 80% of the mass of POA from biomass burning may evaporate when diluted from plume to ambient ~~conditions. Based on partitioning calculations for a range of OA (from 0.01 to $10^5$ ) in Donahue et al. (2006), only the 10 volatility bin exhibits a substantial portions in both gaseous and solid form. The organic components in all other bins are either mostly solid or mostly gaseous. May et al. (2013) also shows that only 10% of biomass burning emissions (Table 2 in May et al. (2013)) is in the 10 bin. Therefore, the potential overestimation arising from not letting these solids in the condition. Theevaporate is no more than 10% (as the lower volatility materials is almost completely solid, and the higher volatility material is almost completely gaseous)~~ for domestic PM2.5 does not include condensables, but this information is not known for the emissions reported to CEIP by other countries.

The performance of this version of the EMEP4UK model simulating a standard suite of gas-phase components and secondary inorganic aerosol PM components is reported in Ots et al. (2016a) comparing with a full year of measurements in London in 2012. In brief, Ots et al. (2016a) reported an NMB of $-1\%$ and $r = 0.79$ for ozone, an NMB of $-32\%$ and $r = 0.78$ for $NO_x$, an NMB of $+6\%$ and $r = 0.73$ for $SO_4^{2-}$, an NMB of $-12\%$ and $r = 0.65$ for $NH_4^+$, and an NMB of $-23\%$ and $r = 0.57$ for $NO_3^-$.

**2.2 Model experiments**

In this study, four different cases were considered. The *Base* case model experiment uses the same emission inventory dataset as Ots et al. (2016a) (i.e. as reported by the NAEI using the splits in Fig. 1), but with a small adjustment in the daily variation in emissions due to temperature, called degree-day factors (Simpson et al., 2012). Recent studies in London have shown